# Analysis of Influential Factors for the Relationship between PM$_{2.5}$ and AOD in Beijing

Caiwang Zheng[1,2], Chuanfeng Zhao[1,2,3]*, Yannian Zhu[1,2,4]*, Yang Wang[1,2], Xiaoqin Shi[1,2], Xiaolin Wu[1,2], Tianmeng Chen[1,2], Fang Wu[1,2], Yanmei Qiu[1,2]

1. State Key Laboratory of Earth Surface Processes and Resource Ecology, and

College of Global Change and Earth System Science, Beijing Normal University,

Beijing 100875, China

2. Joint Center for Global Change Studies, Beijing, 100875, China

10    3. Division of Geological and Planetary Sciences, California Institute of Technology,

Pasadena, CA 91125, USA.

4. Meteorological Institute of Shaanxi Province, Xi'an, China

Correspondence to: Chuanfeng Zhao, czhao@bnu.edu.cn
15                          Yannian Zhu, yannianzhu@gmail.com

**Abstract:** Relationship between aerosol optical depth (AOD) and $PM_{2.5}$ is often investigated in order to obtain surface $PM_{2.5}$ from satellite observation of AOD with a broad area coverage. However, various factors could affect the AOD-$PM_{2.5}$ regressions. Using both ground and satellite observations in Beijing from 2011 to 2015, this study analyzes the influential factors including the aerosol type, relative humidity (RH), planetary boundary layer height (PBLH), wind speed and direction, and the vertical structure of aerosol distribution. The ratio of $PM_{2.5}$ to AOD, which is defined as $\eta$, and the square of their correlation coefficient ($R^2$) have been examined. It shows that $\eta$ varies from 54.32 to 183.14, 87.32 to 104.79, 95.13 to 163.52 and 1.23 to 235.08 $\mu g/m^3$ with aerosol type in spring, summer, fall and winter, respectively. $\eta$ is smaller for scattering-dominant aerosols than for absorbing-dominant aerosols, and smaller for coarse mode aerosols than for fine mode aerosols. Both RH and PBLH affect the $\eta$ value significantly. The higher the RH, the smaller the $\eta$, and the higher the PBLH, the smaller the $\eta$. For AOD and $PM_{2.5}$ data with the correction of RH and PBLH compared to those without, $R^2$ of monthly averaged $PM_{2.5}$ and AOD at 14:00 LT increases from 0.63 to 0.76, and $R^2$ of multi-year averaged $PM_{2.5}$ and AOD by time of day increases from 0.01 to 0.93, 0.24 to 0.84, 0.85 to 0.91 and 0.84 to 0.93 in four seasons respectively. Wind direction is a key factor for the transport and spatial-temporal distribution of aerosols originated from different sources with distinctive physicochemical characteristics. Similar to the variation of AOD and $PM_{2.5}$,

$\eta$ also decreases with the increasing surface wind speed, indicating that the contribution of surface $PM_{2.5}$ concentrations to AOD decreases with surface wind speed. The vertical structure of aerosol exhibits a remarkable change with seasons, with most particles concentrated within about 500 m in summer and within 150 m in winter. Compared to the AOD of the whole atmosphere, AOD below 500 m has a better correlation with $PM_{2.5}$, for which $R^2$ is 0.77. This study suggests that all the above influential factors should be considered when we investigate the AOD-$PM_{2.5}$ relationships.

*Keywords:* $PM_{2.5}$/AOD ratio, aerosol type, relative humidity (RH), planetary boundary layer height (PBLH), wind speed, aerosol vertical distribution.

## 1. Introduction

Atmospheric aerosol, also known as particulate matter, can influence the Earth's climate system by directly and indirectly modifying the incoming solar radiation and outgoing longwave radiation. The direct effect of aerosol on radiation refers to the scattering and absorption of the solar and longwave radiation by aerosol (Charlson et al., 1992; Koren et al., 2004; Lohmann and Feichter, 2005; Qian et al., 2007; Li et al., 2011; Huang et al., 2014; Yang et al., 2016) and the indirect effect of aerosol on radiation is associated with changes in the cloud macro- and micro-physical properties caused by aerosol which can serve as cloud condensation nuclei or ice nuclei (Twomey, 1977; Albrecht, 1989; Kaufman and Fraser, 1997; Feingold, 2003; Garrett et al., 2004; Garrett and Zhao, 2006; Zhao et al., 2012; Hoose and Möhler, 2012; Liu et al., 2012; Zhao and Garrett, 2015). The radiative effect of aerosol is relatively large due to increased emissions of pollution in East Asia (Wang et al., 2010a; Zhuang et al., 2013). Aerosol can also affect the precipitation intensity and pattern by changing cloud microphysical properties (Menon et al., 2002; Qian et al., 2009; Li et al., 2011; Guo et al., 2016a). Meanwhile, aerosol from anthropogenic pollution can cause serious impacts on atmospheric environment and human health by carrying hazardous materials (Pope et al., 2002; Zhang et al., 2007; Samoli et al., 2008; Xu et al., 2013). Thus, it is very important to get accurate information of aerosol properties, such as aerosol optical depth (AOD) and particle matter with size equal or smaller than 2.5

μm aerodynamic diameter ($PM_{2.5}$).

Aerosol properties are often obtained through satellite remote sensing, surface remote sensing, surface and aircraft in-situ observations. Remote sensing observation generally provides the aerosol optical properties such as AOD and aerosol extinction coefficient, but not the aerosol mass or number concentration. Differently, in-situ observation can provide direct measurements of aerosol concentration and $PM_{2.5}$. However, the limited samples for aircraft observation and limited sites for ground-based in-situ observation make it challenging to obtain the $PM_{2.5}$ over many locations, particularly the spatial distribution. Recent studies have proposed methods to estimate the surface $PM_{2.5}$ based on the AOD observations from satellites (van Donkelaar et al., 2006, 2010, 2013; Drury et al., 2008; Wang et al., 2010b; Xin et al., 2016). Although $PM_{2.5}$ from AOD has no high temporal resolution and is not available when it is cloudy or very pollutant, these methods provide the spatial distribution of $PM_{2.5}$ globally or regionally ( Paciorek et al., 2008; Li et al., 2017; Wang et al., 2017).

Many studies have focused on the development of statistical regression models to derive the surface $PM_{2.5}$ from AOD. For example, Guo et al. (2009) for the first time reported the correlation between MODIS AOD and ground-based $PM_{2.5}$ across eastern China based on long-term collocated MODIS AOD and hourly $PM_{2.5}$ measurements from China Atmosphere Watch Network (CAWNET) of Chinese Meteorological Administration. They also discussed the potential influences of planetary boundary layer height (PBLH) and relative humidity (RH) on the correlation between $PM_{2.5}$ and

AOD. van Donkelaar et al. (2010) derived the global $PM_{2.5}$ concentration distribution from satellite-derived AOD using the $PM_{2.5}$/AOD ratios obtained from a global chemical transport model (CTM). Xin et al. (2015) investigated the relationships between $PM_{2.5}$ and AOD over China using observations from the Campaign on atmospheric Aerosol Research-China network during the period from 2012 to 2013.

The relationships between $PM_{2.5}$ and AOD show significant differences over various locations (Corbin et al., 2002; Wang and Christopher, 2003; Hand et al., 2004; Ramachandran, 2005; Kumar et al., 2007; Zhang et al., 2009a; Ma et al., 2014). Some studies (e.g., Ma et al., 2014) have suggested that aerosol types and meteorological conditions can affect the relationship between $PM_{2.5}$ and AOD. However, systematic studies about the influential factors to the relationship between $PM_{2.5}$ and AOD have not been carried out, which are necessary for future derivation of accurate $PM_{2.5}$ from satellite AOD observations. Using both satellite and surface observation of aerosol properties and meteorology variables in Beijing from 2011 to 2015, this study analyzes the influential factors to AOD-$PM_{2.5}$ relationship, which include aerosol type, RH, PBLH, wind speed, and the vertical structure of aerosol distribution.

The paper is organized as follows. Section 2 describes the data and method. Section 3 analyzes the potential influential factors to AOD-$PM_{2.5}$ relationship, and section 4 summarizes the findings.

**2. Data and Method**

**2.1 Data**

The data used in this study include surface $PM_{2.5}$ concentrations and AOD, satellite-based AOD from the moderate-resolution imaging spectroradiometer (MODIS), satellite-based aerosol profiles from the Cloud-Aerosol Lidar and Infrared Pathfinder Satellite Observation (CALIPSO), and meteorology data from China Meteorological Administration (CMA). For comparisons of surface $PM_{2.5}$ and satellite AOD, the hourly surface $PM_{2.5}$ mass concentrations around the satellite overpass time and the instantaneous satellite AOD or aerosol profiles at the grid (5 km resolution for CALIPSO and 10 km resolution for MODIS) closest to the surface site, have been used. For the influential analysis to the surface $PM_{2.5}$ and AERONET AOD relationship, hourly averaged data of $PM_{2.5}$, AOD, and meteorological variables at CMA site (e.g., PBLH, RH and winds) have been adopted. For time without PBLH observations from CMA radiosonde profiles, the 3-hourly PBLH from the European reanalysis data has been interpolated at the grid close to CMA site. The details of these data are described as follows, including the data sources, their spatial and time resolutions, and the data period.

*a. Ground $PM_{2.5}$ Measurements*

The ground-based aerosol observation of $PM_{2.5}$ mass concentrations with hourly time resolution for the period of 2011 to 2015 is obtained from the U.S. Embassy

Beijing site (39.95 °N, 116.47 °E), as reported on the http://www.stateair.net/ website.

The PM$_{2.5}$ mass concentration was measured using the U.S federal reference method.

This method first uses a size selective inlet to remove particles larger than 10 μm, then

takes use of another filter to remove the particles larger than 2.5 μm. The air parcels

before entering by the PM$_{2.5}$ instruments undergo a dry process (RH<35%), which

ensures that all PM$_{2.5}$ mass observations are obtained at dry condition. While this

dataset has not been officially evaluated, a comparison of PM$_{2.5}$ measurements from

U.S Embassy Beijing site and from Beijing Municipal Environmental Protection

(MEP) Bureau site (39.94 °N, 116.46 °E) which are close to each other (1.6 km) in

2014-2016 shows great consistency with correlation coefficient of 0.94 and root mean

square difference of 14.3 ug/m$^3$. Considering that the data measured by U.S. Embassy

Beijing have longer time record, and has been widely used by many studies (Zheng et

al., 2015; Jiang et al., 2015), they are adopted in this study.

### *b. Meteorological Data*

Hourly averaged meteorological parameters at the CMA Beijing site (39.80 °N,

116.47 °E) are provided, including cloud fraction (CF), RH, surface wind speed and

wind direction. The 6-hour total precipitation (TP) observation has also been used in

this study. To eliminate the contamination of cloud and precipitation, data samples

under cloudy (CF≥0.1) or rainy conditions (TP>0) are removed. Same as Yang et al.

(2016), we should note that even with this limitation, some days with few broken

clouds (CF<0.1) still can introduce additional uncertainties to our study. The PBLHs

are extracted from the European Centre for Medium-Range Weather Forecasts (ECMWF) interim reanalysis (ERA-Interim; Dee et al. 2011), with a horizontal resolution of $0.125°\times 0.125°$ and 3-hour temporal resolution. Guo et al. (2016b) have investigated the PBLH in China from January 2011 to July 2015 using both the fine-resolution sounding observations and ECMWF reanalysis data. It was found that the seasonally averaged PBLHs derived from reanalysis are generally in good agreement with those of observations in Beijing. Considering this and that there are only 2 times sounding observations every day, the seasonally averaged ERA PBLHs have been used in this study. We should admit that extra uncertainties could exist due to the distances between the CMA site, U.S. Embassy Beijing site, and ECMWF grid, while they are close to each other.

### c. AERONET measurements

The Aerosol Robotic Network (AERONET) program is a federation of ground-based remote sensing aerosol networks with more than 400 stations globally. At AERONET sites, the CE318 multiband sun-photometer is employed to measure spectral sun irradiance and sky radiances, from which AOD at 550 nm can be derived. The AOD data has been processed into three quality levels: Level 1.0 (unscreened), Level 1.5 (cloud-screened), and Level 2.0 (cloud screened and quality-assured) (Holben et al., 1998). A detailed description about AERONET retrievals is discussed in Holben et al. (1998). In this study, Level 2.0 AOD at 550 nm, SSA at 675 nm and Fine Mode Fraction (FMF) at Beijing AERONET site (39.98 °N, 116.38 °E) with

hourly time resolution are used. It's worth noting that AOD retrieved from AERONET is accurate to within ±0.01 (Dubovik et al., 2000). Note that the AOD retrieved could have the impacts of relative humidity which has not been excluded yet.

*d. CALIOPSO profile products*

CALIPSO is one part of the National Aeronautics and Space Administration (NASA) A-Train, which is a constellation of satellites, tracking in a polar orbit and crossing the equator northbound at about 13:30 local time (LT) (Stephens et al., 2002). To investigate the characteristics of the aerosol vertical distribution, aerosol extinction profiles at 532 nm from Version 3.01 CALIOP Level 2 5 km Aerosol Profile for the

period of 2011 to 2015 are used, which are provided by the CALIOP space borne lidar onboard the CALIPSO satellite (Winker et al., 2007, 2009; Hunt et al., 2009). The horizontal resolution is 5 km, and the vertical resolution varies with altitude. The CALIPSO columnar AOD is the integration of aerosol extinction coefficient with the altitude, which has also been influenced by the relative humidity.

The extraction algorithm of the aerosol profile is shown in Figure 1. First, the overpass time of CALIPSO satellite can be determined according to the geographical location of Beijing site (39.95 °N, 116.47 °E). Second, the aerosol extinction coefficient profiles inside 100 km radius region surrounding the Beijing site are averaged as the final profile result. Third, AOD at each layer is derived as the

integration of the extinction coefficient within that layer. Note that when there are clouds or precipitation, the data are excluded in our analysis. Also, in this process,

low-quality profiles in which *Extinction_Coefficient_Uncertainty_532* (*Sigma_Uncertainty* in Fig. 1) is greater than 99% and *COD* is greater than 0.1 have been excluded.

### *e. MODIS aerosol product*

The MODIS instrument has a global coverage every one to two days with a viewing swath of 2330 km. It is operating on both the Terra and Aqua satellites, of which the overpass time are approximately 10:30 and 13:30 LT, respectively. To compare the AOD from MODIS and CALIPSO (only passes in the afternoon) observation, AOD from Terra (10:30 LT) are not used. Level 2 MODIS aerosol

product data (Collection 5.1) for the period of 2011 to 2015 are obtained from the Level-1 and Atmosphere Archive and Distribution System (LAADS DAAC), of which the spatial resolution at nadir is 10 km$\times$10 km (Levy et al., 2010). The AOD data (MODIS parameter name: Deep_Blue_Aerosol_Optical_Depth_Land) at 550 nm are used in this study, which are only retrieved for daytime, cloud-free and snow/ice-free

conditions with an uncertainty confidence level of ~20%.

### 2.2 Method

### *a. PM$_{2.5}$/AOD ratio*

       AOD represents the total attenuation that aerosols of the whole atmosphere exert on solar radiation. By contrast, PM$_{2.5}$ mass concentrations measured by the ground

monitoring site can only reflect the near-surface air quality condition. Based on the

assumption of linear relationship between AOD (unitless) and $PM_{2.5}$ ($\mu g/m^3$), van

Donkelaar et al. (2010) introduced a conversion factor ($\eta$), which can be defined as:

$$\eta = \frac{PM_{2.5}}{AOD} \tag{1}$$

where $\eta$ ($\mu g/m^3$) indicates the near surface aerosol $PM_{2.5}$ mass concentration per

unit aerosol optical thickness. Its value depends on the aerosol type, aerosol size, RH,

PBLH, and the vertical structure of aerosol distribution. At the same $PM_{2.5}$ mass

concentration, the smaller the AOD, the weaker the extinction capability. Note that the

extinction capability here denotes the aerosol mass extinction coefficient. In other

words, the smaller the $\eta$, the stronger the aerosol extinction capability. Using this

factor, we can study the dependence of AOD-$PM_{2.5}$ relationship (represented by $\eta$) on

different influential factors.

*b. Aerosol Classification Method*

Due to the difference of the sources, aerosols exhibit noticeable differences in

physical and optical properties with respect to the location and season. Fine-mode

fraction (FMF) refers to the fraction of AOD due to fine-mode aerosol particles with

sizes smaller than 1 μm. Angstrom exponent (AE) is exponent for the power law

describing the wavelength dependence of AOD. Using FMF and AE, we can

determine the dominant size mode of aerosol. We can also distinguish absorbing from

non-absorbing aerosols based on measurements of single scattering albedo (SSA),

which is defined as the ratio of the scattering coefficient to the extinction coefficient.

In this study, hourly averaged level 2 inversion products from AERONET at sites in Beijing are used, including FMF, AE and SSA data. Following Lee et al. (2010), aerosol is classified into eight types as follows:

1) Coarse non-absorbing (SSA>0.95, FMF<=0.4 and AE<=0.6)

2) Coarse absorbing (SSA<=0.95, FMF<=0.4 and AE<=0.6)

3) Mixed non-absorbing (SSA>0.95, 0.4<=FMF<0.6 and 0.6<=AE<1.2)

4) Mixed absorbing (SSA<=0.95, 0.4<=FMF<0.6 and 0.6<=AE<1.2)

5) Fine non-absorbing (SSA>0.95, FMF>0.6 and AE>1.2)

6) Fine highly-absorbing (SSA<=0.85, FMF>0.6 and AE>1.2)

7) Fine moderately-absorbing (0.85<=SSA<0.9, FMF>0.6 and AE>1.2)

8) Fine slightly-absorbing (0.9<=SSA<0.95, FMF>0.6 and AE>1.2)

Coarse absorbing and fine absorbing aerosols can be considered as dust and black carbon (BC), respectively. Figure 2 shows the aerosol type classification performed using SSA, FMF and AE from AERONET at sites in Beijing based on the

15 classification method described above. Roughly, the aerosol particles are mainly fine mode slightly absorbing and non-absorbing in summer, and fine mode slightly and moderately absorbing in winter. The coarse mode dust aerosols mainly occur in spring (MAM) and winter (DJF).

**3. Analysis and Results**

**3.1 AOD**

We first evaluate the uncertainties in the satellite observed AOD using the ground measurements from AERONET at the satellite passing time, including those from both MODIS and CALIPSO at 13:30 LT. Based on the satellite overpass time, the corresponding AERONET AODs at time within 30-min frame are compared to MODIS AOD and CALIPSO AOD respectively, which is shown in Figure 3. The correlation between MODIS and AERONET AODs is significant ($R^2 = 0.85$, N = 415), with a slope of 1.32 and an RMS error of 0.23, indicating that MODIS AOD is biased high compared to AERONET AOD. In contrast, the correlation between CALIPSO and AERONET AOD is slightly lower than that between MODIS and AERONET ($R^2 = 0.65$, N = 70), with a slope of 0.78 and a RMS error of 0.31. In general, the CALIPSO AOD is biased low compared to AERONET AOD. The lower correlation of AOD between AERONET and CALIPSO than that between AERONET and MODIS is likely related to the limited data samples for AERONET-CALIPSO AOD comparison, which is also noted by Bibi et al. (2015).

Table 1 further shows the inter-comparison results of AOD between AERONET and MODIS in spring (MAM), summer (JJA), fall (SON) and winter (DJF), which include their seasonal averaged AOD, squared correlation, absolute bias, relative bias and sample number. The absolute bias is calculated as the difference of seasonally

averaged AOD from AERONET and MODIS at the same time; and the relative bias is calculated as the ratio of the absolute bias to the seasonally averaged AERONET AOD. The seasonal averaged AODs are 0.49, 0.61, 0.30 and 0.19 respectively in four seasons for AERONET observations, and 0.66, 0.88, 0.39 and 0.21 for MODIS observations, which are highest in summer but lowest in winter. The corresponding sample numbers are 214, 103, 50 and 48 in four seasons. This seasonal variation pattern is also observed by Yu et al. (2009). MODIS has a large positive bias in spring, summer and fall (36.7, 44.7 and 32.9%), but a smaller positive bias in winter (10.2%). The squared correlations ($R^2$) between MODIS and AERONET in Beijing are 0.81, 0.87, 0.69 and 0.34 in four seasons, of which the corresponding RMSEs are 0.23, 0.29, 0.15 and 0.08, respectively. Low correlation in winter may be caused by the shortage of data samples compared to other seasons. When AOD becomes small, the relative errors in AOD from both MODIS and AERONET become large, which may cause the correlation of AOD between MODIS and AERONET also decrease as demonstrated in Table 1.

Same as Table 1, Table 2 shows the inter-comparison results of AOD between AERONET and CALIPSO in spring, summer, fall and winter. The bias shown in Table 2 is calculated in the same way as that in Table 1. The correlations ($R^2$) between CALIPSO and AERONET AOD are 0.52, 0.48, 0.85 and 0.55 respectively in four seasons. CALIPSO AOD has a positive bias in summer and winter (6.6 and 25.0%), but a negative bias in spring and fall (-5.2 and -14.2%). For all seasons, RMSEs are

less for MODIS than CALIPSO compared to AERONET. As indicated earlier, this is

likely related to the limited data samples for AERONET-CALIPSO AOD comparison.

The results shown in Fig. 2 and Tables 1 and 2 indicate that considerable uncertainties

exist in the satellite observed AOD, introducing up to 45% errors (seasonal biases

5-45%) to the quantification of AOD-PM$_{2.5}$ relationships.

**3.2 Effect of RH and PBLH**

Relative humidity, by affecting the water uptake process of aerosol, can cause a

pronounced change to the aerosol size distribution, chemical composition, and the

extinction characteristics (Liu et al., 2008). The hygroscopic growth factor $f$(RH), can

be defined as the ratio of the aerosol scattering coefficients in ambient with a certain

RH to that in dry air conditions (Li et al., 2014). In this study, $f$(RH) is expressed as

follows in a simple function:

$$f(RH) = \frac{1}{(1 - RH / 100)} \qquad (2)$$

The hygroscopic growth process has a significant contribution to AOD. Since

PM$_{2.5}$ is often measured at a dry condition (<35% in relative humidity), we often need

consider the impacts of relative humidity to AOD in order to get a more reliable

AOD-PM$_{2.5}$ relationship. A dehydration adjustment can be applied to get the dry

condition AOD, which is:

$$AOD_{dry} = \frac{AOD}{f(RH)} \qquad (3)$$

where $AOD_{dry}$ represents the aerosol optical depth with dehydration adjustment.

PBLH influences the vertical profile of particulate matters. In general, the PBLH is dependent on many factors, including meteorological conditions, terrain, sensible heat flux, evaporation and ground roughness (Stull, 1988). Several aircraft observational studies (Liu et al., 2009; Zhang et al., 2009) have found that aerosol particles mainly concentrated within planetary boundary layer (PBL) and that $PM_{2.5}$ mass concentration varies little with height within PBL. Thus, the column integrated $PM_{2.5}$ mass concentration ($PM_{2.5\_column}$) within PBL can be approximated as:

$$PM_{2.5\_column} = PM_{2.5} \times PBLH \qquad (4)$$

In the atmosphere, the RH often increases with height within PBL. This could definitely affect the dehydration adjustment of AOD in Eq. (3). Currently, we only use the surface RH to do the adjustment which could cause that the dry condition AOD is actually somehow overestimated compared to its true value.

Previous studies have shown that aerosols are mainly concentrated within PBL (Guinot et al., 2006; Zhang et al., 2009b). Here, we assume that the column integrated $PM_{2.5}$ within PBL should be comparable to the whole column integrated $PM_{2.5}$. The calculation of column $PM_{2.5}$ mass concentration in Eq. (4) has implied that there are no disconnected aerosol layers and could introduce errors in experimental conditions, which was not considered in this study. Eqs. (3) and (4) imply that for given $PM_{2.5}$, the increase of RH can result in the increase of AOD and the decrease of $\eta$, and that for given AOD, the increase of PBLH can cause the decrease of near-surface $PM_{2.5}$

concentrations and the decrease of $\eta$. Actually, PBLH often correlates with RH, making the separation of PBLH and RH effects challenging. Here, we simply show the effects of both PBLH and RH on the AOD-PM$_{2.5}$ relationship.

Figure 4(a) shows the time series of PBLH (km) and RH (%). In Fig. 4a, the blue bands are for high PBLH and low RH, and the purple bands are for low PBLH and high PBLH, both of which indicate anti-correlated trends between PBLH and RH. Differently, the green (yellow) bands are for low (high) PBLH and low (high) RH, which indicates correlated trends of PBLH and RH. Clearly, there is generally an anti-correlated temporal trend between PBLH and RH. The averaged PBLHs for 2011 to 2015 are 2.56 km, 1.97 km, 1.55 km and 1.32 km with corresponding averaged RHs of 27.58%, 48.73%, 42.78% and 33.05% in MAM, JJA, SON and DJF, respectively. In May, PBLH has the highest value above 2.5 km; and in July, RH has the highest value above 50%. Without considering the variations of sources and sinks, PBLH is negatively correlated with PM$_{2.5}$, and RH is positively correlated with AOD. The anti-correlated trend between PBLH and RH shown in Fig. 4a implies that the effects of PBLH and RH on the AOD-PM$_{2.5}$ relationship could be partially canceled out. However, it is still necessary to consider the effects of PBLH and RH for the study of AOD-PM$_{2.5}$ relationship.

Figure 4(b) shows the temporal variation of monthly averaged AOD and PM$_{2.5}$ at 14:00 LT without any meteorology-based modification to the original observations. It shows a good positive relationship in the time variations of monthly averaged AOD

and $PM_{2.5}$ with a high correlation ($R^2 = 0.63$). Although the temporal trends of AOD

and $PM_{2.5}$ are basically consistent, AOD are considerably higher in MAM and JJA

while $PM_{2.5}$ lower in JJA. That's because in MAM, PBLH is high and the vertical

mixing of aerosol makes near-surface $PM_{2.5}$ concentrations low, while in JJA, RH is

high and the hygroscopic growth of aerosol lead to the increase of AOD. Actually,

PBLH and RH are influenced by the horizontal atmospheric circulation in different

seasons, which contributes to their seasonal variations. Beijing is located in a

mid-latitude East Asian monsoon region. In winter, heavy horizontal winds help the

transportation of aerosol and result in a relatively low AOD, while low PBLH makes

the surface $PM_{2.5}$ relatively high. By contrast, in summer, the high water vapor

transported with the warm air from south makes both AOD and $PM_{2.5}$ relatively high,

while high PBLH makes the surface $PM_{2.5}$ relatively low. These impacts from the

horizontal atmospheric circulation make the seasonal variation of AOD is more

significant than that for surface $PM_{2.5}$, as shown in Fig. 4b.

Figure 4(c) further shows the temporal variation of monthly averaged $AOD_{dry}$

and $PM_{2.5\_column}$ at 14:00 LT which have been adjusted based on Eqs. (3) and (4). Note

that the $AOD_{dry}$ is adjusted based on surface RH using Eqs. (2) and (3) and the vertical

variation of RH has not been considered. As indicated earlier, the $AOD_{dry}$ obtained

here could be somehow overestimated compared to its true value. It shows much

better positive relationship in the temporal variation of monthly average $AOD_{dry}$ and

$PM_{2.5\_column}$, with $R^2$ as 0.76. This result indicates that the corrections for PBLH and

RH are essential for the improvement of the retrieval accuracy of $PM_{2.5}$ from AOD.

Figure 5 compares the diurnal variation of RH and PBLH over four seasons averaged from 2011 to 2015 in Beijing. In terms of seasonal difference, PBLH is the highest in spring (MAM), followed by summer (JJA) and fall (SON), the lowest in winter (DJF), which is consistent with the results found by Guo et al. (2016b). In spring, high PBLH may be associated with the climatologically strongest near-surface wind speed, while in summer, high PBLH could be attributed to the strong solar radiation (Guo et al., 2016b). RH is the highest in summer, followed by fall and winter, the lowest in spring. In terms of diurnal variation, it shows that from 8:00 to 14:00 LT, the solar radiation that surface receives increases, making PBLH rise and RH decrease gradually. It also shows that PBLH at 14:00 LT is the highest and RH at 14:00 LT is the lowest within the whole day. By contrast, from 14:00 to 20:00 LT, the solar radiation that surface receives reduces, thus PBLH goes down and RH increases gradually. PBLH is the lowest and RH is the highest at 23:00 and 2:00 LT respectively within the whole day.

Figure 6 shows the diurnal variation of multi-year (2011-2015) averaged RH and PBLH, AOD and $PM_{2.5}$, $AOD_{dry}$ and $PM_{2.5\_cloumn}$ in four seasons when all four types of measurements are available. The columns represent four seasons of spring, summer, fall and winter and the rows represent different variables. Fig. 6(a1-d1) show that PBLH and RH demonstrate steady increasing and decreasing trends from 6:00 to 17:00 LT, respectively, which are almost the same as their diurnal variation

demonstrated in Fig. 5. As shown in Fig. 6(a2)-(d2), the AOD-PM$_{2.5}$ linear

relationship shows that R$^2$ are 0.1, 0.24, 0.85 and 0.84 in four seasons respectively.

After being corrected for PBLH and RH (Fig. 6(a3-d3)), it shows that R$^2$ values

between AOD$_{dry}$ and PM$_{2.5\_cloumn}$ are 0.93, 0.84, 0.91, 0.93 in four seasons respectively.

These results further indicate that RH and PBLH play essential roles for AOD-PM$_{2.5}$

relationship.

### 3.2 Aerosol type

To study the influence of aerosol type on $\eta$, we analyze the data from 11:00 to

17:00 LT in four seasons respectively. For this time period, the PBLH (RH) has high

(low) values with weak temporal variation, which makes the impacts of PBLH and

RH vary weakly with selected sample time in a season. By doing this, we try to keep a

certain amount of data samples and limit the influence of diurnal variation of RH and

PBLH on $\eta$. The aerosol types can be classified based on the aerosol particle size and

radiative absorptivity, and $\eta$ is a good indicator to the extinction capability of different

aerosol types.

Figure 7 shows the seasonal frequency distribution of aerosol types in four

seasons at Beijing for the period of 2011 to 2015. Dust accounts for 15.4%, 0.4%,

6.4% and 6.9% in spring, summer, fall and winter respectively. Same as that indicated

from Fig. 1, dust aerosol is heavy in spring and winter, particularly in spring. Higher

proportion of dust in spring is mainly associated with the long-range transport from

northwest arid areas (Yan et al., 2015, Tan et al., 2012). Fine mode absorbing aerosols account for 36.5%, 42.6%, 51.1% and 60.3% in four seasons respectively, of which moderately absorbing aerosols account for the highest. Owing to the biomass burning and soot emission generated from heating, the fine mode heavily-absorbing aerosol

percentage is higher in winter than in other seasons, which is 7.7%. The content of fine non-absorbing aerosol is significantly higher in summer and fall than in other two seasons, particularly in summer with a value of 48.4%. As a whole, the aerosol particles in Beijing are primarily fine-mode and absorbing aerosol in terms of particle size and optical property.

Figure 8 presents the variation of $\eta$ with the aerosol type by season in Beijing. Note that there are too few coarse-mode cases in summer and the corresponding $\eta$ is a missing value. $\eta$ generally decreases with particle size, with the smallest value for coarse-mode aerosols and largest value for fine-mode aerosols, and it seems that $\eta$ of non-absorbing aerosols is smaller than absorbing aerosols. Theoretically, aerosol

extinction capacity increases with particle size parameter ($x=2\pi r/\lambda$) and reaches a maximum value when size parameter is around 6. Therefore, for solar visible radiation (such as $\lambda$=500 nm), the extinction capacity for aerosol particles generally increases with size for particles with radius less than 0.5 μm, and then decreases when radius larger than 0.5 μm. Actually, for the wavelength of 550 nm, the extinction

efficiency of fine-mode particles (peak radius ranging from ~0.11 to ~0.33 μm) is stronger than coarse-mode aerosols. Moreover, coarse particles, which may be not

included in $PM_{2.5}$, can contribute a lot to the extinction at wavelengths in the visible, and thus to AOD. This is especially true for dust days dominated by coarse-mode aerosols, of which high AOD is more likely to be due to $PM_{10}$ rather than $PM_{2.5}$. These make the lower $\eta$ for coarse-mode than fine mode aerosol.

Table 3 further compares the AERONET hourly averaged AOD to $PM_{2.5}$ mass concentrations by aerosol type. Coarse Non-absorbing aerosols show the lowest correlation between AOD and $PM_{2.5}$, of which $R^2$ is 0.10. For all kinds of aerosols, the correlation between AOD and $PM_{2.5}$ is relatively lower than that for aerosols with a specific type other than coarse non-absorbing, of which $R^2$ is 0.51 and RMS error is

46.34 μg/m$^3$.

        Figure 9 shows the difference in the relationship between $PM_{2.5}$ and AOD among five different aerosol types by season. The coarse non-absorbing aerosol is too few to be analyzed and thus not shown here. We have also done the linear regression analysis for all types of aerosol which is not shown here, and found that the slopes of the linear

regression functions ($PM_{2.5}=a \times AOD+b$) are 90.16, 56.9, 117.97 and 138.42 in four seasons respectively. The seasonal differences of the slopes are attributed to the effect of PBLH and RH. In summer, high RH brings about the hygroscopic growth of aerosol, thus increasing the extinction capacity of aerosol and then reducing the slope. Moreover, the high PBLH in summer reduces the relative contribution of surface

$PM_{2.5}$ to the columnar AOD and makes a smaller slope value. Differently, in winter, low PBLH value increases the relative contribution of surface $PM_{2.5}$ to the columnar

AOD, thus increasing the slope. The slopes in spring and fall are in between. However, there are large differences in the slope of regression functions among different aerosol types. For absorbing aerosols, the slope roughly increases with decreasing particle size from coarse to mixed particles, with values of about 89, 111 $\mu g/m^3$ in spring, 85, 122 $\mu g/m^3$ in summer, 71, 163 $\mu g/m^3$ in fall, and 44, 143 $\mu g/m^3$ in winter, respectively. The slope is also generally larger for absorbing than non-absorbing aerosol. The slopes for mixed absorbing and non-absorbing aerosol are 111 and 65 $\mu g/m^3$ in spring, 122 and 40 $\mu g/m^3$ in summer, 163 and 109 $\mu g/m^3$ in fall, and 143 and 89 $\mu g/m^3$ in winter. And the slopes for fine absorbing and non-absorbing aerosol are 105 and 76 $\mu g/m^3$ in spring, 74 and 65 $\mu g/m^3$ in summer, 131 and 96 $\mu g/m^3$ in fall, and 158 and 122 $\mu g/m^3$ in winter. Thus, same as shown in Fig. 8, the slope roughly decreases with particle size, with small values for coarse-mode aerosols and large values for fine-mode aerosols in four seasons, and the slope of non-absorbing aerosols is generally smaller than absorbing aerosols.

The findings in this section imply that AOD-PM$_{2.5}$ relationship varies considerably with aerosol types. When we investigate the relationship between PM$_{2.5}$ and AOD, the aerosol types should be carefully considered for study regions.

**3.4 Wind**

This section discusses how wind affects the AOD-PM$_{2.5}$ relationship in two aspects: wind direction and surface wind speed. Surrounded by Hebei province with

severe pollution, Beijing is affected by the long-range transport of aerosol and gas-phase pollutants. The seasonal variation of wind direction changes the transport and spatial-temporal distribution of aerosol and gas-phase pollutants originated from different sources with distinctive physicochemical characteristics, which has a direct

influence on the AOD-PM$_{2.5}$ relationship.

Figure 10 describes the wind rose of Beijing in four seasons for the period from 2011 to 2015. Surface wind speed is mainly distributed in the range of 0 to 9 m/s. Wind direction is mainly southwest in spring and summer, northeast in fall and northwest in winter. There are more windy days in spring and winter. The northwest

wind in spring causes the transport of dust aerosol from gobi and desert regions of China to Beijing. The occurrence frequency of stable weather (v=0 m/s) are 4.2%, 5.8%, 9.2% and 8.3% in spring, summer, fall and winter, respectively. The influence of wind direction to the AOD-PM$_{2.5}$ relationship is often combined with the effect of wind speed. Beijing is surrounded by Hebei province and mountains in the northern

areas. When the winds come from south, Beijing is in the downstream location to the pollution source from Hebei and the pollutants could be further accumulated in Beijing due to the mountain blocking effect. By contrast, when the winds come from north, Beijing is in the upstream region relative to the pollution source in Hebei, and the cold air from north can disperse the air pollutants. As shown in Figure 11, with

similar wind speed, the occurrence rate of heavy air pollution is much higher for cases with winds from the south than from the north. Moreover, the aerosol pollution events

also decrease with increasing wind speed for cases with winds both from the north and the south.

Figure 12 illustrates the relationship between the severity extent of aerosol amount denoted by both AOD and $PM_{2.5}$ and surface wind speed. For good air quality with $PM_{2.5}<50$ μg/m$^3$, the occurrence rate increases with increasing wind speed, ranging from 39.3% (v<=1 m/s) to 92.9% (v>7 m/s). Differently, the occurrence of poor air quality with $PM_{2.5}>150$ μg/m$^3$ ranges from 20.92% (v<=1 m/s) to 0 (v>7 m/s). The weakening of surface wind speed reduces the transport of near-surface aerosol to the outside regions, leading to the build-up and continuance of heavy aerosol pollution condition in Beijing. On the contrary, the increase of surface wind speed, which may be due to the development of weather system like monsoon in Beijing, causes the disperse of aerosol, and then reduction of the heavy aerosol pollution occurrence rate.

Figure 13 describes the variation of averaged AOD, $PM_{2.5}$ and $\eta$ with surface wind speed. Although AOD and $PM_{2.5}$ are basically consistent in the decreasing trend with the increasing surface wind speed, AOD variation is more complicated and less sensitive to surface wind speed. Compared with the $PM_{2.5}$ variation range of 10~110 μg/m$^3$, AOD varies between 0.2 and 0.6. Moreover, there are even cases that AOD increases with wind speed, such as when wind speed is less than 3 m/s. This is likely associated with the fact that the columnar AOD is affected by many factors, and the surface wind speed is just a disturbing term to surface $PM_{2.5}$. Similar to the variation

of AOD and PM$_{2.5}$, $\eta$ also decreases with the increasing surface wind speed, indicating that the contribution of surface PM$_{2.5}$ concentrations to AOD decreases with surface wind speed.

### 3.5 Vertical distribution of aerosol

It has indicated that the relationship between AOD and PM$_{2.5}$ varies with the surface wind speed and the surface aerosol amount. Considering that AOD is the vertical integration of aerosol optical properties, the AOD-PM$_{2.5}$ relationship should vary with the vertical distribution of aerosol. We examine this by using the extinction profiles at 532 nm band from the Version 3.01 CALIOP Level 2 5 km Aerosol Profile product from 2011 to 2015.

Within the atmospheric boundary layer, the main air movement form is the turbulent motion, promoting the vertical exchanges of heat, water vapor, momentum and various kinds of materials including aerosol pollutants. The turbulent energy is generally dependent on both the buoyancy and wind shear, particularly the buoyancy which is highly related to surface downwelling radiation. Obviously, compared to other seasons, the solar radiation received by the surface is more in summer, and the turbulence is stronger, making aerosol transfer to a higher altitude. The seasonal variation of PBLH shown earlier has illustrated this. Associated with the variation of PBLH, the aerosol vertical distribution also varies and further influences the AOD-PM$_{2.5}$ relationship. We next examine the relationship between AOD integrated

from surface to different heights and PM$_{2.5}$ at surface. By defining AOD below a height as the integration of extinction coefficients vertically from surface to that height, the ratio of AOD below a specific height to the total AOD can be determined by CALIPSO vertical profile, which is

$$AOD_H = AOD_{AeronetTotal} \times \frac{AOD_{CalipsoBelowH}}{AOD_{CalipsoTotal}} \quad\quad (5)$$

where $AOD_{AeronetTotal}$ is AOD derived by AERONET, $AOD_{CalipsoTotal}$ is the total AOD from CALIPSO. $AOD_{CalipsoBelowH}$ is AOD below $H$ from CALIPSO, and $AOD_H$ is the AOD below $H$. As shown in Figure 3, the CALIPSO seems underestimate AOD compared to AEORNET. We here treat the AERONET AOD as more reliable or "ground truth" data, and use the CALIPSO vertical profile to scale the AERONET AOD for its vertical distribution.

We here examine four heights, which are 500 m, 1000 m, PBLH and the whole columnar atmosphere that MODIS observes. Note that PBLH varies with time. Figure 14 shows linear relationships between AOD below these four heights and PM$_{2.5}$ at surface. For heights of 500 m, 1000 m, PBLH and the whole atmospheric column, we can see that the correlation between AOD below and surface PM$_{2.5}$ decreases with selected heights, with R$^2$ of 0.77, 0.76, 0.66 and 0.64 respectively. More clearly, the slopes of linear regression lines vary a lot for heights 500 m, 1000 m and PBLH, but much smaller for $H$ above PBLH. This further implies that most of aerosol concentrates within PBL in the atmosphere, and the variation of aerosol vertical distribution could introduce large uncertainties to AOD-PM$_{2.5}$ relationship. PBLH

generally has large diurnal variation and considerable seasonal variation, which is quite different from 500m and 1000m for different seasons. This will inevitably affect the correlation between AOD and PM$_{2.5}$ and its slope.

**4. Summary**

This study analyzes the various factors that affect the AOD-PM$_{2.5}$ relationship qualitatively or quantitatively, including the satellite AOD observation, aerosol type, RH, PBLH, wind direction and speed, and the aerosol vertical distribution. It shows all of these factors can change the AOD-PM$_{2.5}$ relationship, with different contributions. AODs from MODIS and CALIPSO are evaluated against the

AERONET data. The correlation between MODIS and AERONET AOD is significant ($R^2 = 0.85$, N = 415), with a slope of 1.32 and an RMS error of 0.23, indicating that AOD is higher from MODIS than that from AERONET. In contrast, the correlation of AOD between CALIPSO and AERONET is slightly lower ($R^2 = 0.65$, N = 70), with a slope of 0.78 and an RMS error of 0.31.

There are large differences in the seasonal and diurnal variations of PBLH and RH. In Beijing, PBLH is the highest in spring, followed by summer and fall, the lowest in winter, and RH is the highest in summer, followed by fall and winter, the lowest in spring. With the correction of RH and PBLH to AOD, $R^2$ of monthly averaged PM$_{2.5}$ and AOD increases from 0.63 to 0.76 at 14:00 LT, and $R^2$ of

multi-year averaged PM$_{2.5}$ and AOD by time of day increases from 0.01 to 0.93, 0.24 to 0.84, 0.85 to 0.91 and 0.84 to 0.93 in four seasons respectively.

The aerosol particles in Beijing are primarily fine-mode and absorbing aerosols in terms of particle size and optical property. Due to the long-range transport of aerosol from northwest arid areas, dust aerosol is heavy in spring and winter, particularly in spring. It shows that $\eta$ varies with aerosol type, with values ranging from 54.32 to 183.14, 87.32 to 104.79, 95.13 to 163.52 and 1.23 to 235.08 $\mu g/m^3$ in spring, summer, fall and winter, respectively. $\eta$ is generally smaller for scattering-dominant aerosols than for absorbing-dominant aerosols, and smaller for coarse mode aerosols than for fine mode aerosols.

The surface winds significantly affects the occurrence of haze events. With similar wind speed, the occurrence rate of heavy air pollution in Beijing is much higher for cases with winds from the south than from the north. The occurrence rate of good air quality ($PM_{2.5}<50$ $\mu g/m^3$) increases with increasing wind speed, ranging from 39.3% (v<=1 m/s) to 92.9% (v>7 m/s). Differently, the occurrence of poor air quality ($PM_{2.5}>150$ $\mu g/m^3$) ranges from 20.92% (v<=1 m/s) to 0 (v>7 m/s). It shows that $\eta$ decreases with the increasing surface wind speed, indicating that the contribution of surface $PM_{2.5}$ concentrations to AOD decreases with surface wind speed.

The vertical structure of aerosol distribution exhibits a remarkable change with seasons, which could also contribute a lot to the AOD-$PM_{2.5}$ relationship. This study shows that aerosols mainly concentrate within about 500 m height in summer, while concentrate within the surface layer of around 150 m height in winter in Beijing. Compared to the AOD of the whole atmosphere, AOD below 500 m has a better

correlation with $PM_{2.5}$, of which $R^2$ is 0.77 and RMSE is 38.6 $\mu g/m^3$.

With these findings, we need consider at least the impacts of PBLH, RH, Wind speed and wind direction, and use the AOD within PBL heights to build up better AOD-$PM_{2.5}$ relationship. The impacts of these influential factors have been
investigated while an optimal empirical AOD-$PM_{2.5}$ relationship scheme has not been reached, which definitely need further study in future.

**Acknowledgements**

This work was supported by the National Natural Science Foundation of China ( NSFC: grant 41575143), the Ministry of Science and Technology of China (grants 20
13CB955802), the China "1000 Plan" young scholar program, and the Chinese Program for New Century Excellent Talents in University (NCET). Sincerest thanks to the AERONET, MODIS and CALIPSO teams for their datasets. The CALIPSO data were obtained from the NASA Langley Research Center Atmospheric Science Data Center. Special thanks to the U.S. Embassy and CMA providing the $PM_{2.5}$ data and
meteorological data respectively.

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

**Table 1.** Comparison of AERONET and MODIS AOD by season and over all seasons.

| Season | AERONET mean AOD | MODIS mean AOD | $R^2$ | Bias | Bias% | RMSE | $N$ |
|--------|------------------|----------------|-------|------|-------|------|-----|
| Spring | 0.49 | 0.66 | 0.81 | 0.18 | 36.7 | 0.23 | 214 |
| Summer | 0.61 | 0.88 | 0.87 | 0.27 | 44.7 | 0.29 | 103 |
| Fall | 0.30 | 0.39 | 0.69 | 0.10 | 32.9 | 0.15 | 50 |
| Winter | 0.19 | 0.21 | 0.34 | 0.02 | 10.2 | 0.08 | 48 |
| All | 0.46 | 0.63 | 0.85 | 0.17 | 37.8 | 0.23 | 415 |

*Note:* Bias% is defined as $100 \times$ (MODIS AOD-AERONET AOD)/AERONET AOD (Green et al., 2009). RMSE is the root mean squared prediction error ($\mu g/m^3$).Period for comparison is 2011–2015.

**Table 2.** Comparison of AERONET and CALIPSO AOD by season and over all seasons

| Season | AERONET mean AOD | CALIPSO mean AOD | $R^2$ | Bias | Bias% | RMSE | $N$ |
|--------|------------------|------------------|-------|------|-------|------|-----|
| Spring | 0.44 | 0.42 | 0.52 | -0.02 | -5.2 | 0.33 | 21 |
| Summer | 0.53 | 0.57 | 0.47 | 0.04 | 6.6 | 0.32 | 16 |
| Fall | 0.95 | 0.81 | 0.85 | -0.14 | -14.2 | 0.34 | 12 |
| Winter | 0.42 | 0.53 | 0.55 | 0.11 | 25.0 | 0.27 | 21 |
| All | 0.54 | 0.55 | 0.65 | 0.01 | 1.7 | 0.31 | 70 |

**Table 3.** Correlations between AOD and PM$_{2.5}$ mass by dominant aerosol type and for all aerosols

| Dominant Aerosol type | R$^2$ | RMSE (μg/m$^3$) | N |
|---|---|---|---|
| Coarse Absorbing | 0.56 | 27.07 | 480 |
| Mixed Absorbing | 0.67 | 36.44 | 1383 |
| Fine Absorbing | 0.53 | 48.06 | 2143 |
| Coarse Non-absorbing | 0.10 | 44.51 | 56 |
| Mixed Non-absorbing | 0.61 | 44.05 | 234 |
| Fine Non-absorbing | 0.58 | 40.19 | 434 |
| All | 0.51 | 46.34 | 4728 |

## Figure Captions

**Figure 1.** Flow chart of deriving aerosol vertical profile from CALIPSO data.

**Figure 2.** The aerosol classification scheme in four seasons from 2011 to 2015 using AE, SSA and FMF data from AERONET at sites in Beijing. The scatter plots of different colors are the distribution of aerosol types with different physic-optics characteristics in four seasons of spring (MAM), summer (JJA), fall (SON) and winter (DJF).

**Figure 3.** Scatter plots of AERONET AOD vs. MODIS AOD (a), and AERONET AOD vs. CALIPSO AOD (b) for the period of 2011 to 2015 in Beijing. The solid red line represents the best fit line using the linear regression.

**Figure 4.** Comparison of monthly averaged RH and PBLH (a), AOD and $PM_{2.5}$ (b), $AOD_{dry}$ and $PM_{2.5\_column}$ (c) at 14:00 LT for the period of 2011 to 2015 in Beijing. The blue, purple, green and yellow bands in (a) are for high PBLH and low RH, low PBLH and high PBLH, low PBLH and low RH, high PBLH and high RH, respectively.

**Figure 5.** Diurnal variations of multi-year (2011-2015) averaged RH and PBLH over four seasons (MAM, JJA, SON, and DJF) in Beijing.

**Figure 6.** Comparison of multi-year (2011-2015) averaged RH and PBLH (a1~d1), AOD and $PM_{2.5}$ (a2~d2), $AOD_{dry}$ and $PM_{2.5\_column}$ (a3~d3) by time of day in different seasons. The columns represent four seasons (MAM, JJA, SON, and DJF) and the rows represent three different variables.

**Figure 7.** The frequency distribution of aerosol types over four seasons (MAM, JJA, SON, and DJF) for the period of 2011 to 2015 in Beijing.

**Figure 8.** The variation of $\eta$ with the aerosol type in four seasons (MAM, JJA, SON, and DJF) for the period of 2011 to 2015.

**Figure 9.** Scatter plots between AERONET AOD and $PM_{2.5}$ concentrations in four different seasons (MAM, JJA, SON, and DJF) for five different types of aerosols. The first to 5[th] columns represent the aerosol types of coarse absorbing, mixed absorbing, fine absorbing, mixed non-absorbing, and fine non-absorbing, respectively. The colors also represent different aerosol types. The rows represent four seasons. The solid black line represents the best fit line using linear

regression

**Figure 10.** Wind rose of Beijing in four seasons (MAM, JJA, SON, and DJF) for the period of 2011 to 2015.

**Figure 11.** The relative distribution of $PM_{2.5}$ within different value ranges at Beijing for different surface wind speed in different wind direction.

**Figure 12.** The relative distribution of AOD (upper panel) and $PM_{2.5}$ (lower panel) within different value ranges at Beijing for different surface wind speed ranges from 2011 to 2015. v and N represent the wind speed and samples respectively. The colors represent the value ranges of AOD (upper panel) and $PM_{2.5}$ (lower panel).

**Figure 13.** Variation of averaged AOD, $PM_{2.5}$ (left panel) and $\eta$ (right panel) with the surface wind speed. For the right panel, the solid red line and the dashed black line represent the best fitting using linear regression and quadratic regression, respectively.

**Figure 14.** Scatter plots of stratified AOD vs. $PM_{2.5}$ concentrations. It shows the relationship between (a) AOD below 500m, (b) AOD below 1000m, (c) AOD below PBL and (d) AOD of the whole atmosphere and $PM_{2.5}$ concentrations. The solid red line is the linear fitting regression line and the dashed red lines represent the 95% confidence interval of the linear fitting regression line.

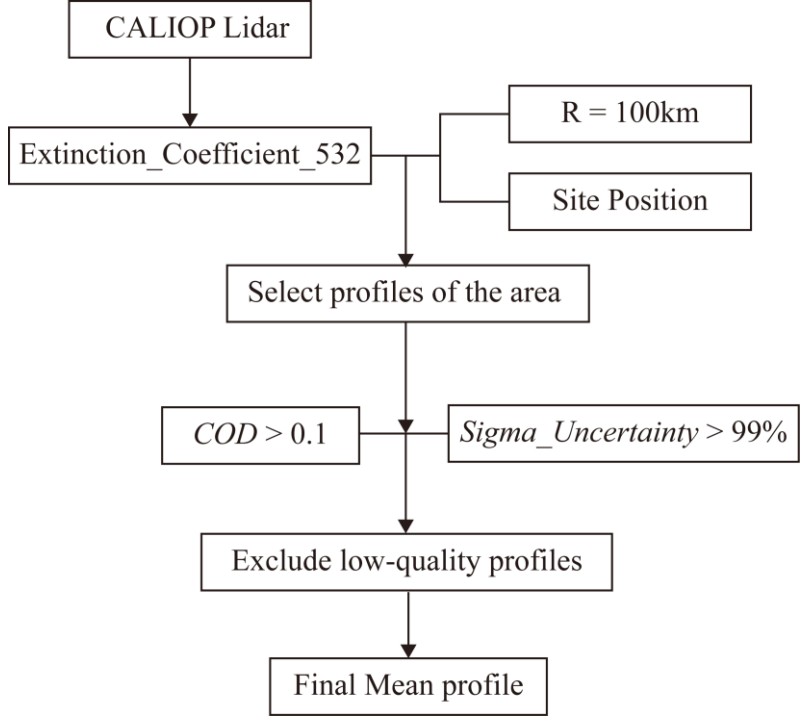

**Figure 1.** Flow chart of deriving aerosol vertical profile from CALIPSO data.

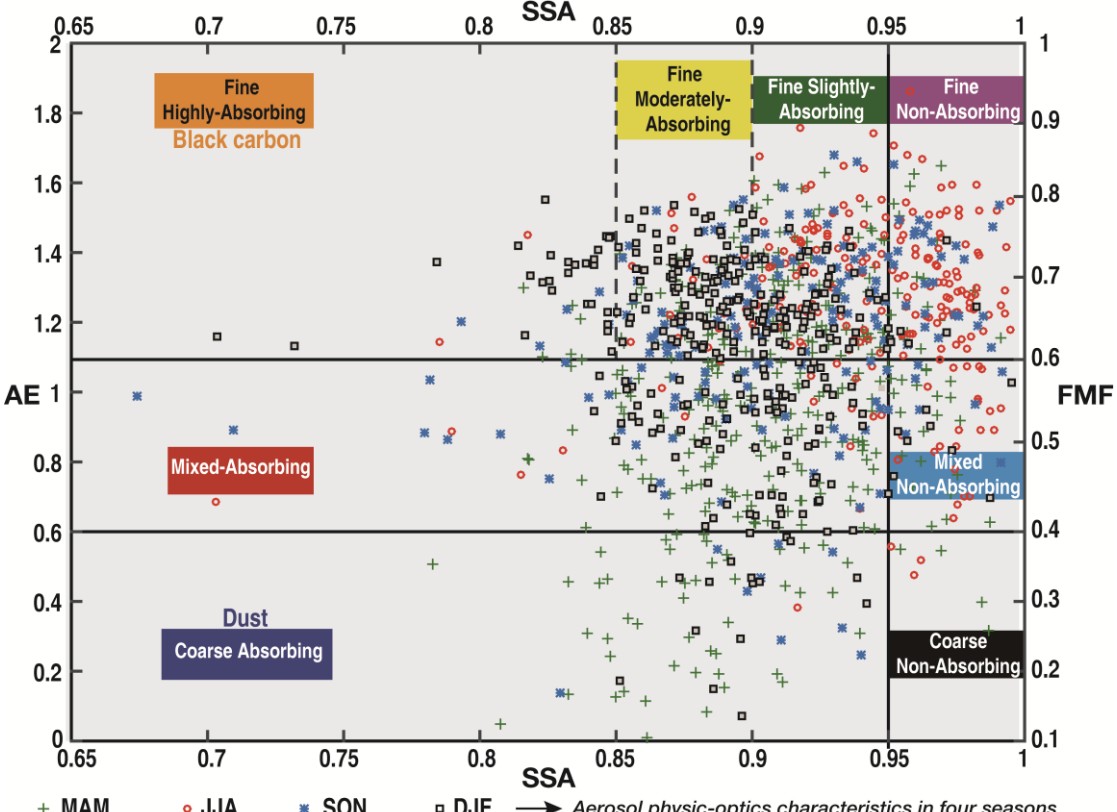

**Figure 2.** The aerosol classification scheme in four seasons from 2011 to 2015 using AE, SSA and FMF data from AERONET at sites in Beijing. The scatter plots of different colors are the distribution of aerosol types with different physic-optics characteristics in four seasons of spring (MAM), summer (JJA), fall (SON) and winter (DJF).

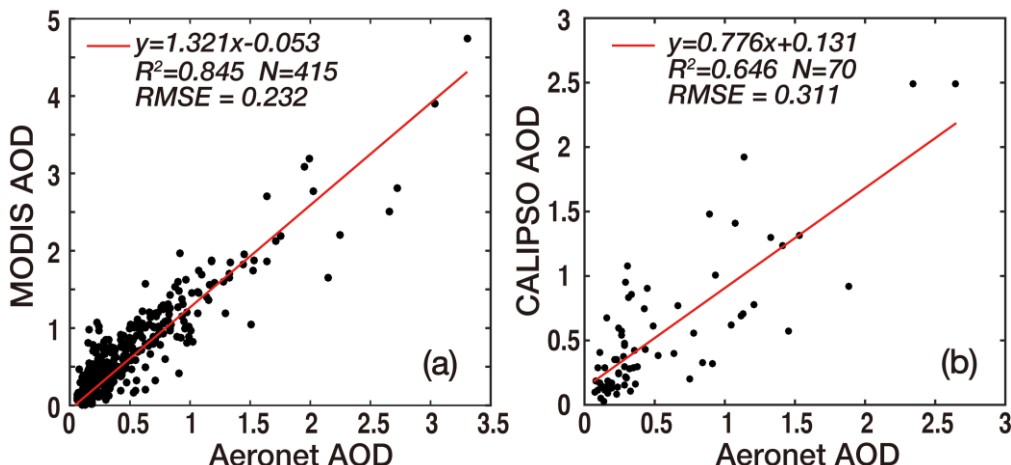

**Figure 3.** Scatter plots of AERONET AOD vs. MODIS AOD (a), and AERONET AOD vs. CALIPSO AOD (b) for the period of 2011 to 2015 in Beijing. The solid red line represents the best fit line using the linear regression.

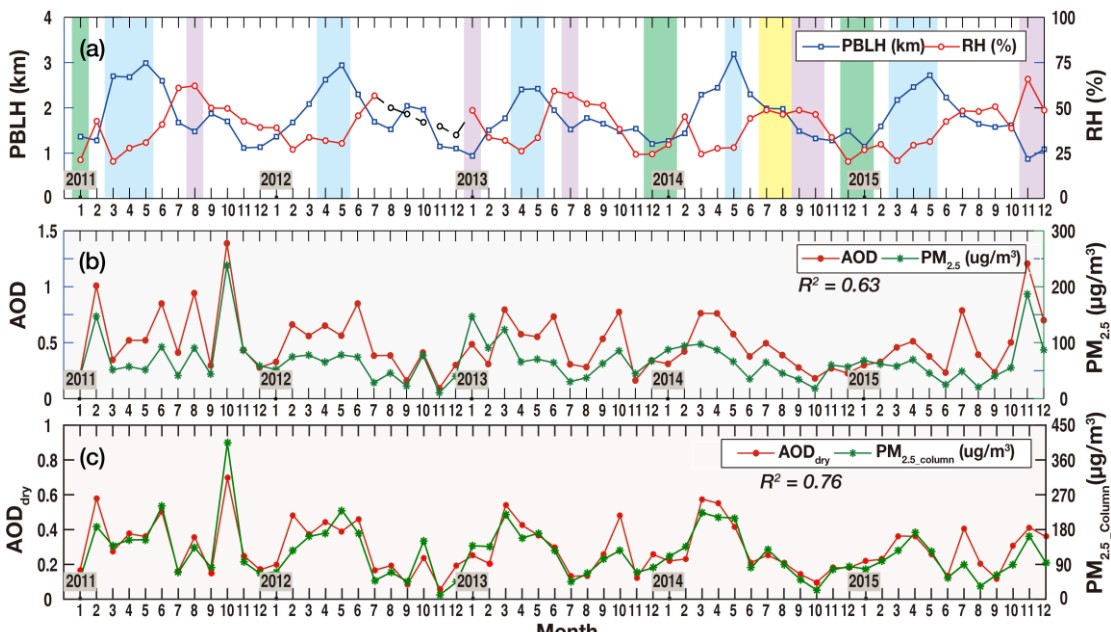

**Figure 4.** Comparison of monthly averaged RH and PBLH (a), AOD and PM$_{2.5}$ (b), AOD$_{dry}$ and PM$_{2.5\_column}$ (c) at 14:00 LT for the period of 2011 to 2015 in Beijing. The blue, purple, green and yellow bands in (a) are for high PBLH and low RH, low PBLH and high PBLH, low PBLH and low RH, high PBLH and high RH, respectively.

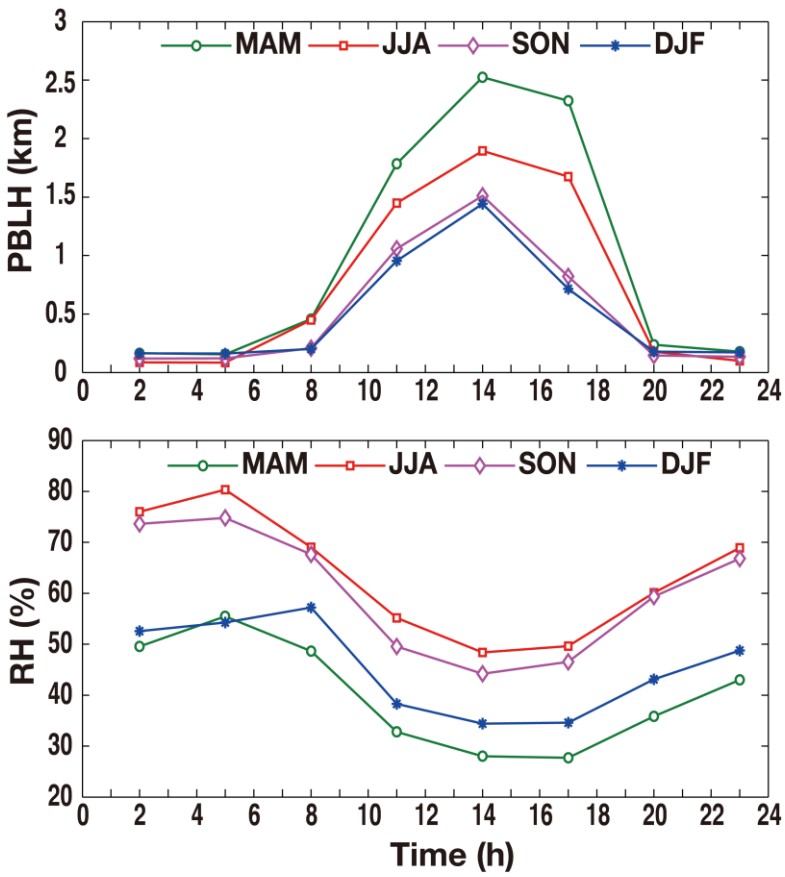

**Figure 5.** Diurnal variations of multi-year (2011-2015) averaged RH and PBLH over four seasons (MAM, JJA, SON, and DJF) in Beijing.

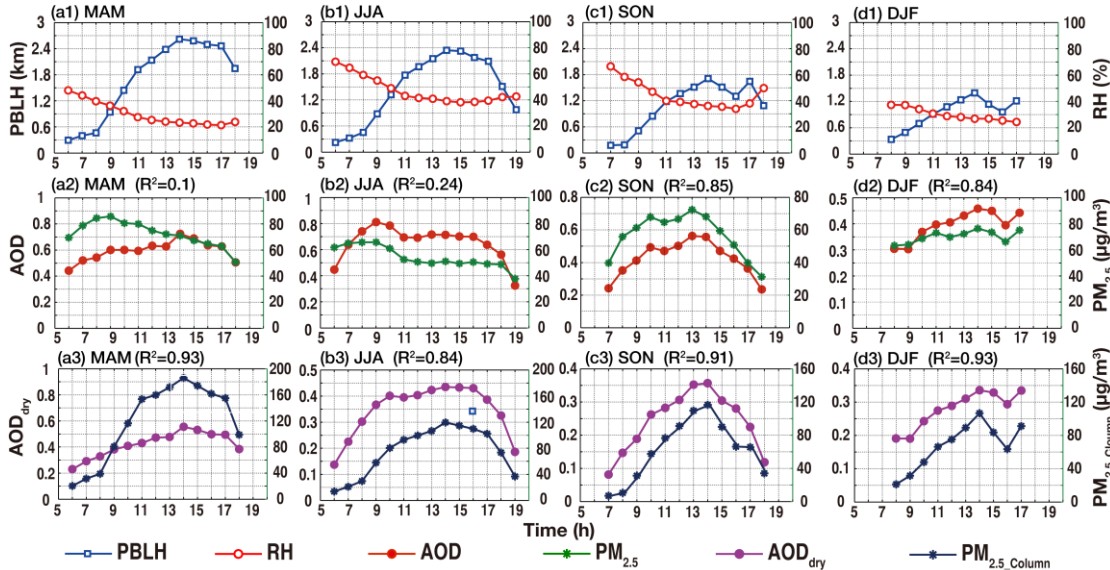

**Figure 6.** Comparison of multi-year (2011-2015) averaged RH and PBLH (a1~d1), AOD and PM$_{2.5}$ (a2~d2), AOD$_{dry}$ and PM$_{2.5\_column}$ (a3~d3) by time of day in different seasons (MAM, JJA, SON, and DJF). The columns represent four seasons and the rows represent three different variables.

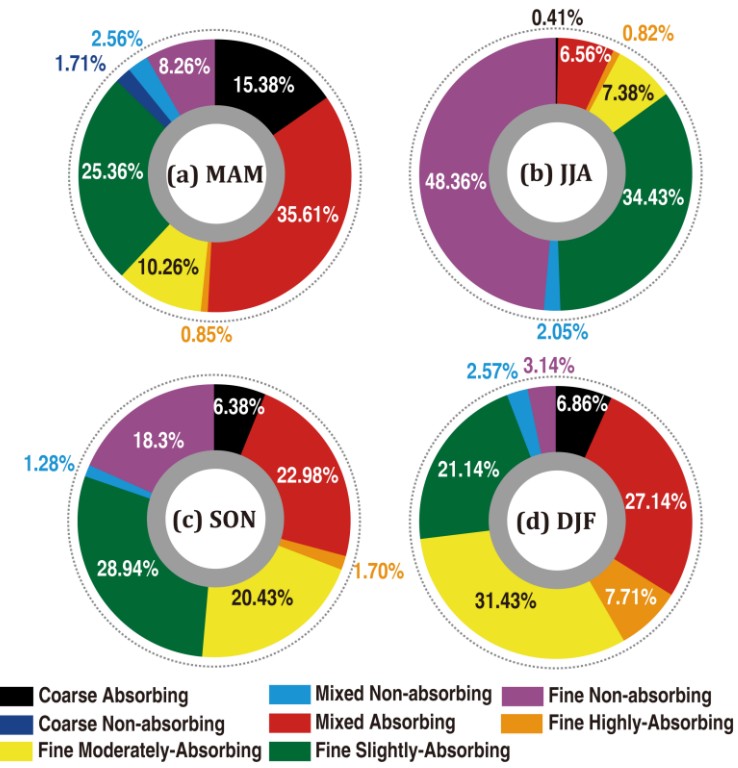

**Figure 7.** The frequency distribution of aerosol types over four seasons (MAM, JJA, SON, and DJF) for the period of 2011 to 2015 in Beijing.

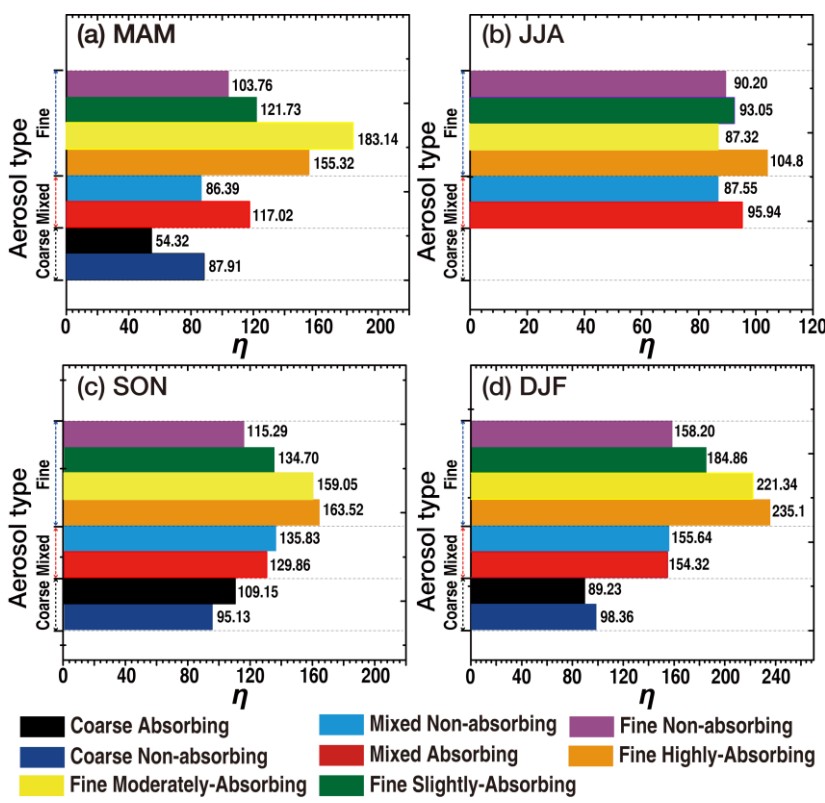

**Figure 8.** The variation of $\eta$ with the aerosol type in four seasons (MAM, JJA, SON, and DJF) for the period of 2011 to 2015.

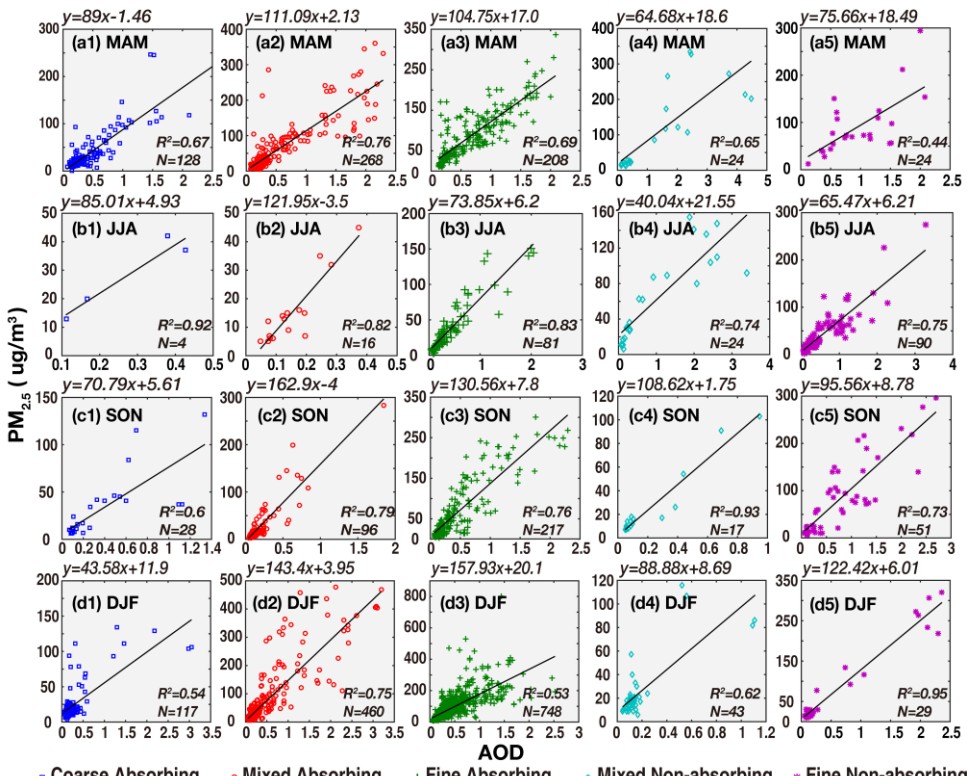

**Figure 9.** Scatter plots between AERONET AOD and PM$_{2.5}$ concentrations in four different seasons (MAM, JJA, SON, and DJF) for five different types of aerosols. The first to 5$^{th}$ columns represent the aerosol types of coarse absorbing, mixed absorbing, fine absorbing, mixed non-absorbing, and fine non-absorbing, respectively. The colors also represent different aerosol types. The rows represent four seasons. The solid black line represents the best fit line using linear regression.

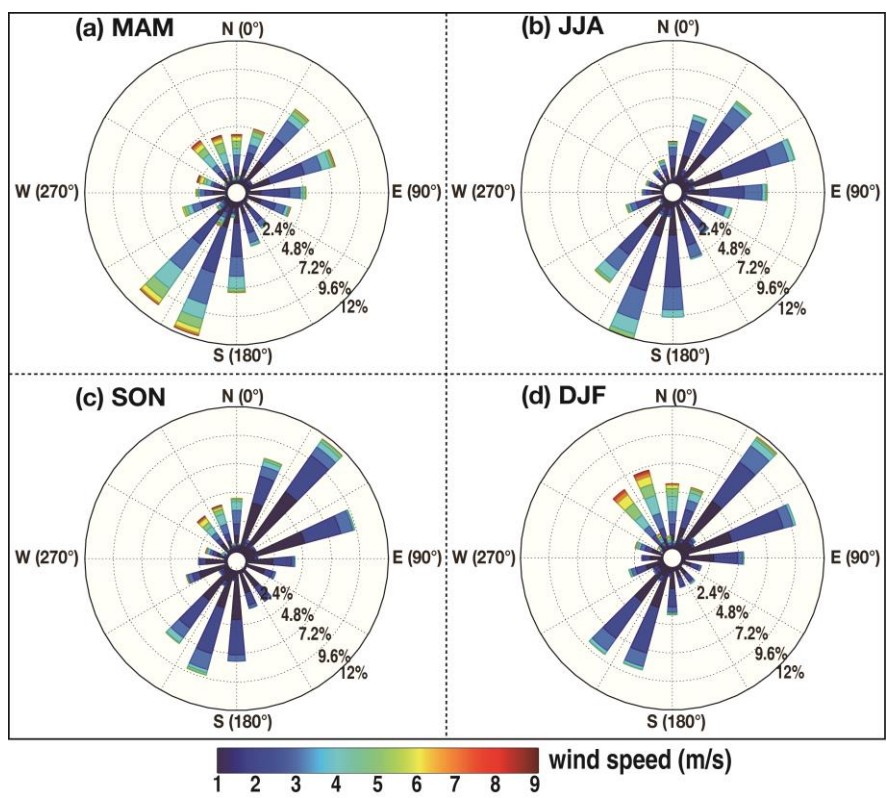

**Figure 10.** Wind rose of Beijing in four seasons (MAM, JJA, SON, and DJF) for the
period of 2011 to 2015

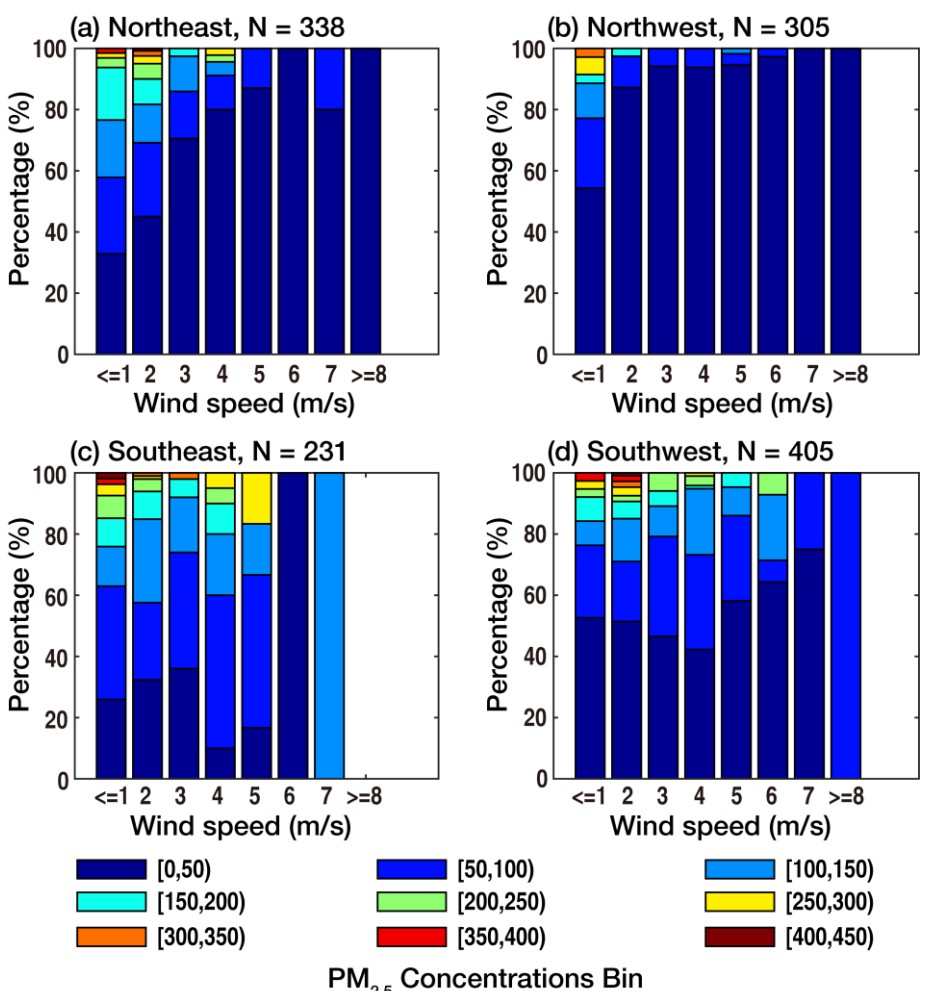

**Figure 11.** The relative distribution of PM$_{2.5}$ within different value ranges at Beijing for different surface wind speed in different wind direction.

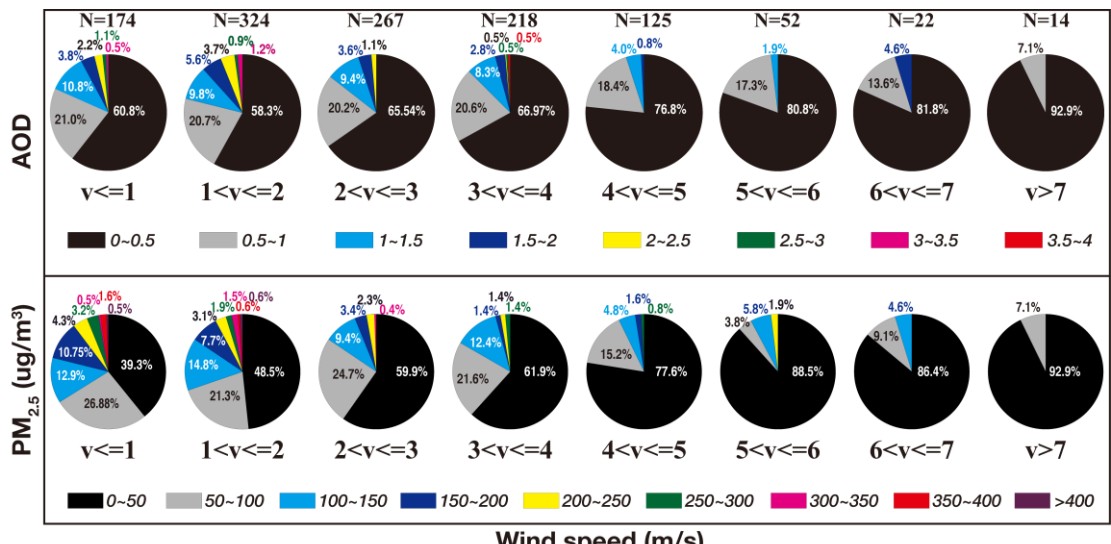

**Figure 12.** The relative distribution of AOD (upper panel) and PM$_{2.5}$ (lower panel) within different value ranges at Beijing for different surface wind speed ranges from 2011 to 2015. v and N represent the wind speed and samples respectively. The colors represent the value ranges of AOD (upper panel) and PM$_{2.5}$ (lower panel).

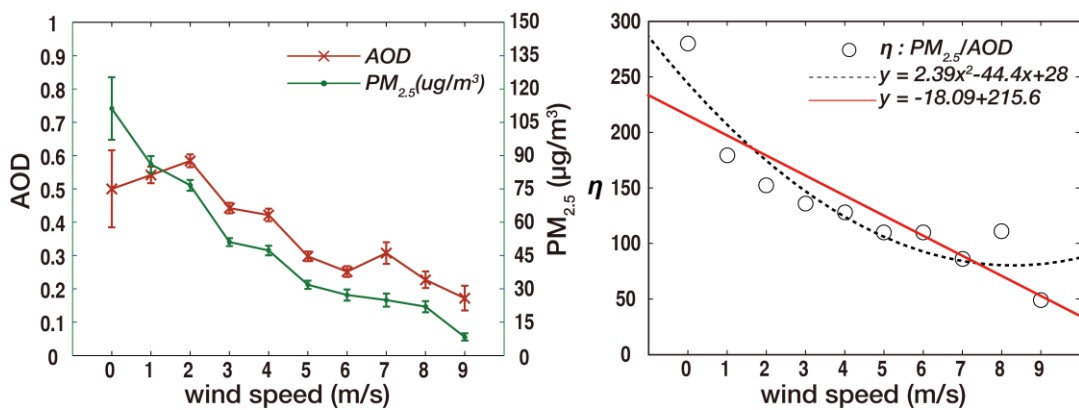

**Figure 13.** Variation of averaged AOD, PM$_{2.5}$ (left panel) and $\eta$ (right panel) with the surface wind speed. For the right panel, the solid red line and the dashed black line represent the best fitting using linear regression and quadratic regression, respectively.

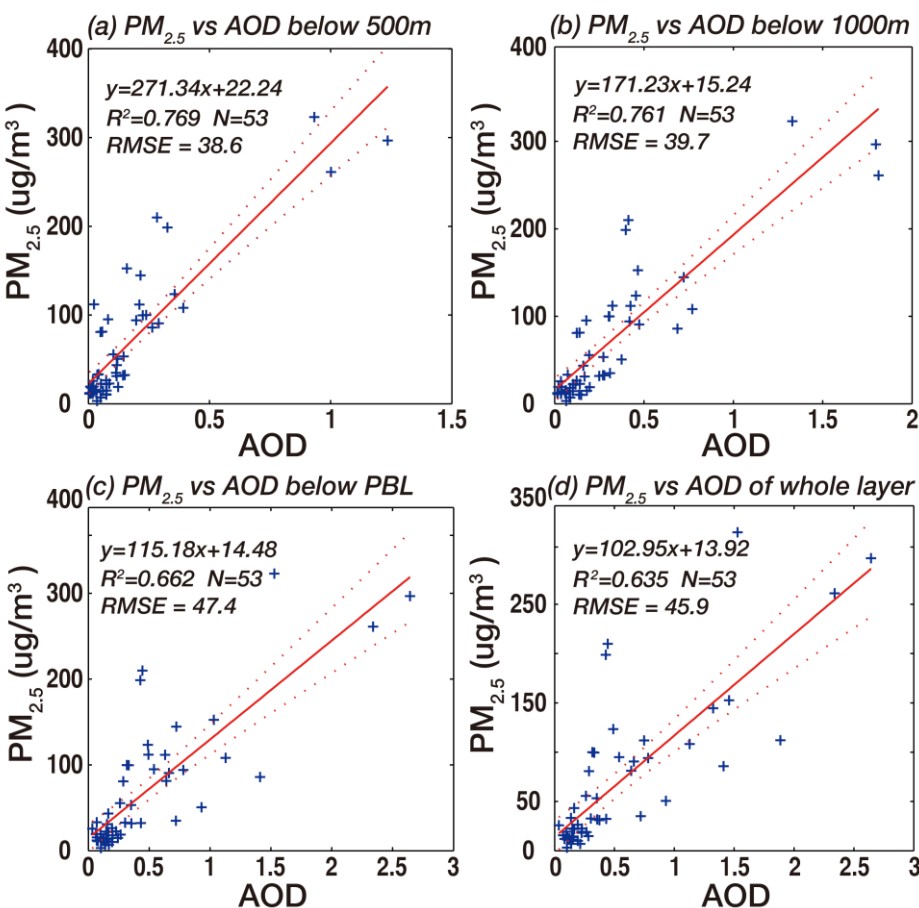

**Figure 14.** Scatter plots of stratified AOD vs. PM$_{2.5}$ concentrations. It shows the relationship between (a) AOD below 500m, (b) AOD below 1000m, (c) AOD below PBL and (d) AOD of the whole atmosphere and PM$_{2.5}$ concentrations. The solid red line is the linear fitting regression line and the dashed red lines represent the 95% confidence interval of the linear fitting regression line.

