# Peer review of "Analysis of Influential Factors for the Relationship between $PM_{2.5}$ and AOD in Beijing"

_Atmospheric Chemistry and Physics, 2016_

## Referee Comment (RC1) · Anonymous Referee #1 · 7 Mar 2017

The relationship between satellites observed aerosol optical depth to PM2.5 at surface is important to monitor and quantify the air pollution at ground from satellite observations which is relatively more independent and has less artificial errors than ground-based measurements. However, the physical connection between AOD and PM2.5 is relatively indirect and weak thus can be affected by multiple factors. This study utilized most existing state-of-art measurements and observations of aerosol and meteorology parameters to investigate the impacts from aerosol type, relative humidity (RH), the height of boundary layer (PBLH), wind speed, wind direction and the vertical distribution of aerosol on the relationship of AOD-PM2.5 in Beijing, China. This investigation is comprehensive, timing and with high scientific significance given the very serious situation of air pollution in China. I suggest to accept it for publication in ACP after minor revisions and addressing the following questions.

[Figure]

P13, L1: Why the correlation of AOD between AERONET and CALIPSO is lower than that between AERONET and MODIS, any interpretations?

P13,L7: Any explanation about the reason of "AOD becomes small, it seems that the correlation of AOD between MODIS and AERONET also decreases."

P13, L15-17: a lot of information contained in Table 1 and 2 needs more in-depth discussions.

P15, L1: add "for given PM2.5" before "the increase of RH can result in . . . .."

P15, L2: add "for given AOD" before "the increase of PBLH can cause . . . .. ."

P15,L18-21: This part of discussion need consider the impacts from horizontal atmosphere circulation in different seasons.

P16, L20: the first row of Figure 6 seems redundant with Figure 5.

P17,L4: "0.01" should be "0.1"

P18, L14: The aerosol extinction capacity should "decrease" with increasing particle size.

P19, L6: better be consistent, using "extinction capacity" ?

P19,L9-14: The information in Figure 9 is not thoroughly revealed yet: how will the size, the absorption/scattering capability impact$\eta$? Can the conclusion be hold in all seasons?

P20, L1 & L3: "aerosol" → "aerosol and gas-phase pollutants". Because second order aerosol can form from gas-phase air pollutants.

P20, L20 – P21, L5: The effect of wind speed should work with wind direction, depending on the relative location of Beijing to the pollution source (i.e. upstream or downstream).

P21,L8-9: Why AOD increase with increasing wind speed slower than 3m/s?

P22,L20: This is not true based on Figure 13: significant AOD were contributed by aerosols located above 500m altitude. The contribution from lower layer (<500m) is smaller comparing to that from upper layer (>500m).

P23, L1-4: Discussion here should be combined with the discussion of PBLH.

---

## Referee Comment (RC2) · Anonymous Referee #3 · 4 Jun 2017

Review of Zheng et al., "Analysis of influential factors for the rlationship between PM2.5 and AOD in Beijing" submitted for publication in ACP, May 2017.

**General comments**

The authors investigate the relationship between AOD and PM2.5 using 4 years of data at a single site in Beijing. In particular, they investigate the influence of factors such as PBLH, RH, wind speed and direction and aerosol type. The rationale for this study is to explain the variability of the PM2.5/AOD relationship because of the possible application of satellite observations of AOD for PM2.5 monitoring.

In this study the authors use AOD from MODIS-Aqua (L2, C5.1 from the DB algorithm) and AERONET (L2 AOD, AE, FMF and SSA; these parameters were used to classify aerosol type) and aerosol profiles from CALIOP, together with hourly PM2.5 data from the U.S. Department of State which are freely available from a website. However, no information is provided on how these data were measured and what parameter is reported (dry/wet aerosol). Also, a disclaimer on this website (http://www.stateair.net/web/assets/USDOS_AQDataFilesFactSheet.pdf) states that the data have not been validated or quality assured. Hence the authors should clarify what they have done to do this, and why these data were selected for their study over other possibly available PM2.5 data. The PM2.5/AOD ratio was analyzed using 3-hourly meteorological data provided by the CMA (CF, TP, RH, wspd and WD. PBLH was extracted from ECMWF interim re-analysis data.

This is a very comprehensive suite of experimental data. All of these are obtained at different times and with different spatial scales and, for ground-based measurements, at different locations. Hence the data sets need to be colocated in both space and time before any analysis can be undertaken. Although locations indicated in section 2 are within 1 degree, an inhomogeneous megacity environment, both as regards surface characteristics and aerosol sources, may result in substantial spatial variation of the aerosol characteristics. These points need to be addressed in the MS.

Overall, the manuscript shows the influence of various factor on the PM2.5/AOD relationship, but presentation lacks clarity and many things are left unexplained as detailed in my comments below. I miss clear conclusions (only a summary is provided at the end) and how the findings can be combined to provide a relationship between PM2.5 and AOD, if that would be possible even for a single site.

In general, the figures would be more interesting if they were easier to read, in some cases the text is too small. Furthermore, although a comparison is made between MODIS and AERONET data, in the main part of the MS only AERONET and CALIPSO data are sued. Hence I do not understand why MODIS is included.

**Detailed comments**

p4, lines 2-3    specify modifying: scattering and absorption ?

5, 3    usually it's not the extinction coefficient that is provided but the AOD, i.e. the integrated extinction over the whole column

5, 10    spelling van Donkelaar (also elsewhere in the MS)

5, 11-13    these statements need some references

6,18    what do you mean with 'and so on'? Either specify or remove

8, 11    Was AERONET version 2 or the new version 3 used?

9, 8    No, AOD is not extinction multiplied by layer depth, but the integral of the extinction over the whole layer. This is different when extinction is not constant with height, as is usually the case.

9, 20    why was MODIS C5.1 used and not the newer C6?

10, 2    why was Deep Blue selected and not dark target? Or the merged DB/DT AOD product from C6?

10, 14    is it the optical thickness per unit mass concentration, or inversely the mass concentration per unit AOD, as eq 1 says?

10, 16-19         it is not clear what extinction capability means: extinction per unit mass, or mass extinction coefficient? Or extinction efficiency? Or ….? Is it really true that for the same PM2.5, the extinction is weaker for the same AOD? Or is it more complicated and does size distribution and RH have an influence?

11, 3    discrepancies or differences?

11,5    FMF is the fraction of the AOD due to fine particles (smaller than 1 micrometer); AE is exponent for the power law describing the wavelength dependence of the AOD

11, 9    SSA is the ratio of the scattering coefficient to the extinction coefficient (=scattering + absorption)

11, 12    replace According to with Following Lee et al. (2010)

12, 2    what do you mean with 'since the high percentage of '?

12,6    replace ' the method' with 'the classification method'

13,6    what does it mean when MODIS correlated best, considering the bias, i.e. considering that MODIS AOD is too high?

Tables 1 and 2: what parameter is listed in the first 2 columns? How is bias calculated?

14, 1    suggest to change to: the hygroscopic growth factor f/RH) is defined as the ratio  … ; is it really the hygroscopic growth factor? This factor would relate to particle size, which indeed is one of the underlying factors, in addition to refractive index, influencing scattering; however, this article is about AOD, which is scattering and thus the scattering enhancement factor should be considered, see for instance Zieger et al., (2015) (Low hygroscopic scattering enhancement of boreal aerosol and the implications for a columnar optical closure study, Atmos. Chem. Phys., 15, 7247-7267, doi:10.5194/acp-15-7247-2015, 2015) and references therein for a discussion on the subject and the f(RH) for different aerosol types in Europe.

14,5    how was eq 2 derived? Has this eq been compared with measured hygroscopic growth curves, or better scattering enhancement?

14, 6-8 I don't understand this explanation since both the MODIS and AERONET AOD are derived in ambient conditions, and for the same air mass when well colocated. Hence hygroscopic growth cannot be the physical reason for larger MODIS AOD than AERONET AOD, since these should be the same. More likely these are retrieval errors. Since in this MS CALIPSO AOD is the main source for analysis, should a similar discussion be made for CALIPSO?

14, 13    what is the consequence of this assumption? Uniform mixing would imply that dry aerosol particles, water vapour and potential temperature are well-mixed, but RH would in that case increase with height and thus all aerosol parameters that vary with RH. In addition, often a scale height is used to account for an aerosol gradient.

14, 19    this assumption implies that there are no disconnected aerosol layers and thus may introduce errors in experimental conditions where these may occur, as revealed by lidar.

15, 2-3  I would agree that an increase of PBLH could result in the decrease of surface PM2.5, in the absence of sources and sinks for PM2.5, but that does not follow from eqs 3 and 4, unless PM2.5column would be assumed constant.

15, 7    opposite trend, suggest to replace with anti-correlated; what are the colour bands in Fig 4a?

15, 11  I assume that this is a typo and fig 4a is meant, but still don't understand the sentence

16, top para: here AOD dry is plotted where AODdry was obtained using the correction factor given by eq 2, which has to be explained. However, as mentioned above, RH is not constant with height, and hence f(RH) also varies with height. How was this accounted for? It seems that the good correlation between AODdry and PM2.5 is somewhat fortuitous. Does this good agreement lead ot the conclusion that PM2.5 was measured dry (at low RH)? See also the discussions on vertical variation of aerosol profiles later in this MS.

Figs 5 and 6 both show the diurnal variation of PBLH and RH, for different seasons averaged over many years, , for the whole day in Fig. 5, and for 5-20 in Figure 6 (a-d). I would expect that the curves for each season are exactly the same for the overlapping time periods. Could the authors explain the differences?

17,4-8   why is a linear relationship expected between AOD and PM2.5, while the authors showed already in the above that there is not a good correlation, as also shown in Figs 6a2-d2? Why are these plots shown, why not only the corrected AODdry, since the necessity of that correction was discussed above? Furthermore, Figs 6a3-d3 do not show the R2, they show time series which show that part of the time AODdry and PM2-5 trace each other, but at other times not; furthermore, these plots show the offset, i.e. AODdry much larger than PM2.5, which is in contrast to what was shown in Figure 4c, where the time series are much closer. Could the authors explain the differences? The PM2.5 scales in Figs 4c and Fig 6 a3-d3 are much different but plotting on the same scales would make the discrepancy even larger.

17, 10-12        why only this narrow time frame (11-17 LT)? why does this limit the diurnal and seasonal variation on PM2.5/AOD?

17, 13  Is the classification based on the scheme described on p. 11? Please refer to that scheme, and if not provide a reference. In Figure 7 are 8 aerosol types are shown whereas on p. 11 there are only 6, so I suspect that the some other classification than that on p. 11 is used

18, 5    what is 'heavily non-absorbing aerosols'? This type neither occurs in Fig 7 nor on p. 11.

Figures 7 and 8 present the same aerosol types, but with a different colour scheme. It would be much easier to read when the colour schemes are the same. Please change.

18, 14  extinction capacity? On p. 10 extinction capability was used. Is that different? I don't understand either of these terms, see my comment on p. 10.

18, Fig 8 and Table 3 discussion: rather than only providing correlation coefficients, the authors should discuss the reasons for the observed differences. There are two fundamental reasons for the variation of the

PM2.5 / AOD ratio: PM2.5 does not include coarse particles while do they do contribute to the extinction at wavelengths in the visible, and thus to AOD. In addition, the extinction efficiency peaks for a particle size which depends on the ratio between particle size and the wavelength.

19, 1-5   Figure 9 is too small to read the numbers, but it is clear that there are large differences in the slopes from one aerosol type to another, as also stated on line 9. Hence I don't understand where the numbers given on line 3 come from.

19, 6   why does high RH increase the extinction efficiency? Since the refractive index of water is lower than that of most other materials, would the condensation of water vapour lower the extinction efficiency?

19, 10   Similar as on line 3, I do not understand where these numbers come from. I cannot find them in Fig 9

19, 12   Figure 4 does not show that the slope decreases with particle size.

19, 15   this conclusion is clear, if only looking at Figure 9. However, I do not understand how the discussion in this section supports this conclusion.

20, 13-15   this is a very surprising conclusion, regarding the location of Beijing with near-by mountains blocking the circulation, pollution advected form the south and clean air from the north, as well as the transport of desert dust aerosol in the spring. What leads to this conclusion? I don't see it in Figure 10. And also the first 5 lines on top of p. 2 contradict this conclusion.

20, Fig 11 does not show haze occurrence but AOD and PM2.5.

22, 2 and fig 13: what is actually plotted: the axis annotation reads AOD, but AOD thus far was the column-integrated extinction, i.e. independent of height. May be extinction is plotted? Should these profiles reflect that the PBLH varies with season, and thus also the aerosol vertical distribution vary with season? The strong decrease with height contradicts the assumption of a uniform mixing (p. 14).

22, 9   as for Fig 13, I don't understand what is plotted here

22, 18-21: same comments as for the top of the page. I don't understand what is shown here, and hence it is hard to comment. I also don't understand why in summer, with peak AOD at a height of 516 m, is concluded that most aerosol is below that height. Integration of the profiles below and above 500 m would give a number supporting this conclusion.

23, eq 5   This eq needs some more explanation. Apparently the authors assume here that CALIPSO AOD underestimates the true value (fig 3)? And therefore they scale to AERONET AOD?

23, 15   how is top of atmosphere defined?

Figure 15 would be easier to understand when the AOD was plotted on the same scale. Note that, PM2.5 is plotted vs AOD, not AOD vs PM2.5 as the caption says.

24, 1-3   this conclusion is clear and follows form Figure 13, and hence Fig 15 is obsolete, including the explanation on p.23; it also shows that the uniform-mixing assumption on p 14. is not valid for aerosol.

---

## Author Comment (AC1)

Response to Reviewer #1

We thank the reviewer for the valuable comments and suggestions that have helped us improve the paper. Our detailed responses (**Bold**) to the reviewers' questions and comments (*Italic*) are listed below.

*P13, L1: Why the correlation of AOD between AERONET and CALIPSO is lower than that between AERONET and MODIS, any interpretations?*

**This is a good point. The correlation of AOD between AERONET and CALIPSO (r$^2$ =0.646, N = 70) is found lower than that between AERONET and MODIS (r$^2$ =0.845, N = 415). One reason is the insufficient data samples for AERONET-CALIPSO AOD comparison, which is also noted by Bibi et al. (2015). Another likely reason is that all these observations have their uncertainties which could make the correlations hardly expected. We have added this into our manuscript in P14 L6-8:** "The lower correlation of AOD between AERONET and CALIPSO than that between AERONET and MODIS is likely related to the limited data samples for AERONET-CALIPSO AOD comparison, which is also noted by Bibi et al. (2015)**".**

**Bibi, H., Alam, K., Chishtie, F., et al.: Intercomparison of MODIS, MISR, OMI, and CALIPSO aerosol optical depth retrievals for four locations on the Indo-Gangetic plains and validation against AERONET data, Atmospheric Environment, 111, 113-126, 2015.**

*P13, L7: Any explanation about the reason of "AOD becomes small, it seems that the correlation of AOD between MODIS and AERONET also decreases."*

**AOD retrieved from AERONET are accurate to within ±0.01 (Dubovik et al., 2000). When AOD becomes small, the relative errors in AOD from both MODIS and AERONET become large, which may be the reason that causes the correlation of AOD between MODIS and AERONET also decrease. We have added this discussion into Pg15 L3-6:** "When AOD becomes small, the relative errors in AOD from both MODIS and AERONET become large, which may cause the correlation of AOD between MODIS and AERONET also decrease as demonstrated in Table 1.**".**

*P13, L15-17: a lot of information contained in Table 1 and 2 needs more in-depth discussions.*

**We agree with the reviewer and made further descriptions and discussions in the manuscript, including**

**(1) The seasonal variations of AOD from different observations and the likely reason for their differences, which are added in P14 L9-19:** "Table 1 further shows the inter-comparison results of AOD between AERONET and MODIS in spring (MAM), summer (JJA), fall (SON) and winter (DJF), which include their seasonal averaged AOD, squared correlation, absolute bias, relative bias and sample number. The absolute bias is calculated as the difference of seasonally averaged AOD from

AERONET and MODIS at the same time; and the relative bias is calculated as the ratio of the absolute bias to the seasonally averaged AERONET AOD. The seasonal averaged AOD are 0.49, 0.61, 0.30 and 0.19 respectively in four seasons for AERONET observations, and 0.66, 0.88, 0.39 and 0.21 for MODIS observations, which are highest in summer but lowest in winter. The corresponding sample numbers are 214, 103, 50 and 48 in four seasons. This seasonal variation pattern is also observed by Yu et al. (2009).".

**(2) The correlation coefficient and RMSE error between MODIS and AERONET AOD observations, which are added into Pg14 L20-Pg15 L3: "**The squared correlation ($R^2$) between MODIS and AERONET in Beijing are 0.81, 0.87, 0.69 and 0.34 in four seasons, of which the corresponding RMSEs are 0.23, 0.29, 0.15 and 0.08. Low correlation in winter may be caused by the shortage of data samples compared to other seasons.".

**(3) An explanation for the lower RMSE value between MODIS and AERONET than that between CALIPSO and AERONET, which are in P15 L13-15: "**For all seasons, RMSE are less for MODIS than CALIPSO compared to AERONET. As indicated earlier, this is likely related to the limited data samples for AERONET-CALIPSO AOD comparison".

*P15, L1: add "for given PM2.5" before "the increase of RH can result in : : :.."*
**Corrected**

*P15, L2: add "for given AOD" before "the increase of PBLH can cause : : :: : :"*
**Corrected**

*P15, L18-21: This part of discussion need consider the impacts from horizontal atmosphere circulation in different seasons.*
**We highly appreciate this valuable comment. We have added the discussion about the impacts of horizontal atmospheric circulation in Pg19 L1-10: "**Actually, PBLH and RH are influenced by the horizontal atmospheric circulation in different seasons, which contributes to the seasonal variations of $PM_{2.5}$ and AOD. Beijing is located in a mid-latitude East Asian monsoon region. In winter, heavy horizontal winds help the transportation of aerosols and result in a relatively low AOD, while low PBLH makes the surface $PM_{2.5}$ relatively high. By contrast, in summer, the high water vapor transported with the warm air from south makes both AOD and $PM_{2.5}$ relatively high, while high PBLH makes the surface $PM_{2.5}$ relatively low. These impacts from the horizontal atmospheric circulation make the seasonal variation of AOD is more significant than that for surface $PM_{2.5}$, as shown in Fig. 4b.".

*P16, L20: the first row of Figure 6 seems redundant with Figure 5.*
**This is a good point. Actually, what Figure 6 shows in the first row is the diurnal variation of multi-year averaged RH and PBLH when all measurements of RH, PBLH, AOD, PM2.5 are available. By contrast, Figure 5 shows the variation when RH and PBLH are available. Even so, the first row in Figure 6 does show**

similar diurnal trends as those shown in Figure 6. We have modified the description to clarify these in P20 L12-14: "Figure 6 shows the diurnal variation of multi-year (2011-2015) averaged RH and PBLH, AOD and PM2.5, AODdry and PM2.5_column in four seasons when all four types of measurements are available." and indicated the similar seasonal variation in P18 L15-17: "Fig. 6(a1-d1) show that PBLH and RH demonstrate a steady increase and decrease trend from 6:00 to 17:00 LT, respectively, which is almost the same as their diurnal variation demonstrated in Fig. 5.".

*P17, L4: "0.01" should be "0.1"*
**Corrected.**

*P18, L14: The aerosol extinction capacity should "decrease" with increasing particle size.*

**The reviewer proposed a good question. In general, the aerosol extinction capacity depends on the size parameter ($2\pi*r/\lambda$). For size parameter between 0 and about 6, the extinction capacity increases with the size parameter. For solar visible radiation (such as $\lambda$=500 nm), the extinction capacity for aerosol particles generally increases with size for particles with radius less than 0.5 $\mu$m, which lies within fine and coarse modes. We have modified our description in P22 L11-Pg23 L1:** "Theoretically, aerosol extinction capacity increases with particle size parameter ($x=2\pi r/\lambda$) and reaches a maximum value when size parameter is around 6. Therefore, for solar visible radiation (such as $\lambda$=500 nm), the extinction capacity for aerosol particles generally increases with size for particles with radius less than 0.5 $\mu$m, and then decreases when radius larger than 0.5 $\mu$m. Actually, for the wavelength of 550 nm, the extinction efficiency of fine-mode particles (peak radius ranging from ~0.11 to ~0.33 $\mu$m) is stronger than coarse-mode aerosols. Moreover, coarse particles, which may be not included in $PM_{2.5}$, can contribute a lot to the extinction at wavelengths in the visible, and thus to AOD. This is especially true for dust days dominated by coarse-mode aerosols, of which high AOD is more likely to be due to $PM_{10}$ rather than $PM_{2.5}$. These make the lower $\eta$ for coarse-mode than fine mode aerosol." .

*P19, L6: better be consistent, using "extinction capacity"?*
**Corrected**

*P19, L9-14: The information in Figure 9 is not thoroughly revealed yet: how will the size, the absorption/scattering capability impact? Can the conclusion be hold in all seasons?*

**We highly appreciate the reviewer's valuable comments here. We have added further analysis in depth and found the corresponding impacts which are shown in Pg23 L20-Pg24 L13:** "However, there are large differences in the slope of regression functions among different aerosol types. For absorbing aerosols, the slope roughly decreases with increasing particle size from coarse, mixed to fine particles, with values of about 89, 111, 104 $\mu$g/m$^3$ in spring, 85, 122, 74 $\mu$g/m$^3$ in summer, 71, 163, 131 $\mu$g/m$^3$

in fall, and 44, 143, 158 μg/m$^3$ in winter. The slope is also generally larger for absorbing than non-absorbing aerosol. The slopes for mixed absorbing and non-absorbing aerosol are 111 and 65 μg/m$^3$ in spring, 122 and 40 μg/m$^3$ in summer, 163 and 109 μg/m$^3$ in fall, and 143 and 89 μg/m$^3$ in winter. And the slopes for fine absorbing and non-absorbing aerosol are 105 and 76 μg/m3 in spring, 74 and 65 μg/m$^3$ in summer, 131 and 96 μg/m$^3$ in fall, and 158 and 122 μg/m$^3$ in winter. Thus, same as shown in Fig. 8, the slope roughly decreases with particle size, with small values for coarse-mode aerosols and large values for fine-mode aerosols in four seasons, and the slope of non-absorbing aerosols is generally smaller than absorbing aerosols.**".**

*P20, L1 & L3: "aerosol" to "aerosol and gas-phase pollutants". Because second order aerosol can form from gas-phase air pollutants.*
**Modified as suggested in Pg 24 L19-Pg25 L3.**

*P20, L20 – P21, L5: The effect of wind speed should work with wind direction, depending on the relative location of Beijing to the pollution source (i.e. upstream or downstream).*
**We highly agree with the reviewer's comment. Beijing is surrounded by Hebei province and mountains in the northern areas. When the winds come from south, Beijing is in the downstream location to the pollution source from Hebei and the pollutants could be further accumulated in Beijing due to the mountain blocking effect. By contrast, when the winds come from north, Beijing is in the upstream region relative to the pollution source in Hebei, and the cold air from north can disperse the air pollutants. Considering the wind speed and direction, the occurrence rate of heavy air pollution is higher for south wind than north wind, which is shown in Fig R1 (also current Figure 11). With the consideration that this study concentrates on the effect of the influential factors to the AOD-PM$_{2.5}$ relationship rather than the air pollution, we added corresponding descriptions in Pg 25 L11-Pg26 L2**: "Different from the wind speed which will be analyzed in Figs. 12 and 13, the influence of wind direction to the AOD-PM$_{2.5}$ relationship is often combined with the effect of wind speed. Beijing is surrounded by Hebei province and mountains in the northern areas. When the winds come from south, Beijing is in the downstream location to the pollution source from Hebei and the pollutants could be further accumulated in Beijing due to the mountain blocking effect. By contrast, when the winds come from north, Beijing is in the upstream region relative to the pollution source in Hebei, and the cold air from north can disperse the air pollutants. As shown in Figure 11, with similar wind speed, the occurrence rate of heavy air pollution is much higher for cases with winds from the south than from the north. Moreover, the aerosol pollution events also decrease with increasing wind speed for cases with winds both from the north and the south.".

[Figure]

Fig. R1. The relative distribution of PM$_{2.5}$ within different value ranges at Beijing for different surface wind speed in different wind direction

*P21, L8-9: Why AOD increase with increasing wind speed slower than 3m/s?*

**This is a good question. As indicated in the paper, the AOD variation is more complicated and less sensitive to surface wind speed, since the columnar AOD is affected by many factors and the surface wind speed is just a disturbing term to surface PM$_{2.5}$. Actually Fig 13(a) even shows that AOD increases with increasing wind speed when it is less than 3m/s, while the variation of the averaged AOD with the wind speed is small (0.50, 0.54 and 0.58 for wind speed 0, 1, 2m/s respectively as shown in Figure R2). Even though, we still can see a decreasing trend of AOD with wind speed in general. We have added corresponding discussions in Pg26 L15-Pg27 L1:** "Although AOD and PM$_{2.5}$ are basically consistent in the decreasing trend with the increasing surface wind speed, AOD variation is more complicated and less sensitive to surface wind speed. Compared with the PM$_{2.5}$ variation range of 10~110 μg/m$^3$, the variation range of AOD is between 0.2 and 0.6. Moreover, there are even cases that AOD increases with wind speed, such as when wind speed is less than 3 m/s. This is likely associated with the fact that the columnar AOD is affected by many factors, and the surface wind speed is just a disturbing term to surface PM$_{2.5}$.".

[Figure]

Fig. R2. The AOD histogram distribution in different wind speed (v=0, 1, 2m/s)

Response to Reviewer #2

    We thank the reviewer for the detailed and valuable comments and suggestions that have helped us improve the paper. Our detailed responses (**Bold**) to the reviewers' questions and comments (*Italic*) are listed below.

*Review of Zheng et al., "Analysis of influential factors for the relationship between $PM_{2.5}$ and AOD in Beijing" submitted for publication in ACP, May 2017.*

*General comments*
*The authors investigate the relationship between AOD and $PM_{2.5}$ using 5 years of data at a single site in Beijing. In particular, they investigate the influence of factors such as PBLH, RH, wind speed and direction and aerosol type. The rationale for this study is to explain the variability of the $PM_{2.5}$/AOD relationship because of the possible application of satellite observations of AOD for $PM_{2.5}$ monitoring.*
*In this study the authors use AOD from MODIS-Aqua (L2, C5.1 from the DB algorithm) and AERONET (L2 AOD, AE, FMF and SSA; these parameters were used to classify aerosol type) and aerosol profiles from CALIOP, together with hourly $PM_{2.5}$ data from the U.S. Department of State which are freely available from a website. However, no information is provided on how these data were measured and what parameter is reported (dry/wet aerosol). Also, a disclaimer on this website (http://www.stateair.net/web/assets/USDOS_AQDataFilesFactSheet.pdf) states that the data have not been validated or quality assured. Hence the authors should clarify what they have done to do this, and why these data were selected for their study over other possibly available $PM_{2.5}$ data.*

    **We highly appreciate the detailed and invaluable comments of the reviewer for our study.**
    **Regarding the data measurements and parameters used in this study, it has been briefly described in section 2.1 and a little more information about how they were measurements are added into Pg 7 line 6-18:** "The $PM_{2.5}$ mass concentration was measured using the U.S federal reference method. This method first uses a size selective inlet to remove particles larger than 10 μm, then takes use of another filter to remove the particles larger than 2.5 μm. The air parcels before entering the $PM_{2.5}$ instruments undergo a dry process (RH<35%), which ensure that all $PM_{2.5}$ observations are obtained at dry condition. While this dataset has not been officially evaluated, a comparison of $PM_{2.5}$ measurements from U.S Department of State and from Beijing Municipal Environmental Protection Bureau at sites close to each other (1.6 km) in 2014 – 2016 shows great consistency with correlation coefficient of 0.94 and root mean square difference of 14.3 ug/m$^3$. Considering that the data measured by U.S. Department of State have longer time record, and have been widely used by many studies (Zheng et al., 2015; Jiang et al., 2015), they are adopted in this study.", **Pg 9 Line 4-5**: "Note that the AOD retrieved could have the

impacts of relative humidity which has not been excluded yet", **Pg 9 line 16** "…
which has also been influenced by the relative humidity", a**nd Pg 9 line 15-21, and
Pg10 line 15-17**: "…which is only retrieved for daytime, cloud-free and snow/ice-free
conditions with an uncertainty confidence level of ~20%".

**Regarding the use and data quality of PM$_{2.5}$ from the U.S. Department of State,
the reviewer proposed a good question. We have noticed the disclaimer which
states that this data has not been validated or quality assured before our study.
There are two reasons why we use the hourly PM$_{2.5}$ data from the U.S.
Department of State.**

1) **The PM$_{2.5}$ observations at this station have a much longer record than those
we can find at other sites, such as the sites operated by the Beijing Municipal
Environmental Protection Bureau (MEP). The initial time for the
observations by the US Department of State was on April 8, 2008 while the
initial time for the data we can get from MEP was on May, 2013. We would
like to use the observations as long as possible in our study. Actually, the time
period of our study is from 2011 to 2015. Therefore, the hourly PM$_{2.5}$ data
from the U.S. Department of State is better for the needs of our study.**

2) **This data has been widely used by many studies (e.g., Zheng et al., 2015;
Jiang et al., 2015) and quickly evaluated in our early analysis (not provided in
the paper). We did a quick examination by comparing the PM$_{2.5}$ observations
from U.S Department of State and from Beijing Municipal Environmental
Protection Bureau at sites close to each other (1.6 km) for the period from
2014 to 2016. As shown in Figure R1, the two observations show great
consistency with each other: the correlation coefficient is as high as 0.94 and
root mean square difference is ~14 µg/m$^3$. These two datasets are basically
consistent. Data samples under rainy conditions (TP>0) have been removed
to eliminate the influence of precipitation. So we believe that the U.S.
Department of state data is reliable. This information has also been added
into Pg7 L11-18:** "While this dataset has not been officially evaluated, a
comparison of PM$_{2.5}$ measurements from U.S Department of State and from
Beijing Municipal Environmental Protection Bureau at sites close to each other
(1.6 km) in 2014 – 2016 shows great consistency with correlation coefficient of
0.94 and root mean square difference of 14.3 ug/m3. Considering that Ttheis data
measured by U.S. Department of State haves relatively longer time record, and
has been widely used by many studies (Zheng et al., 2015; Jiang et al., 2015), they
are adopted in this study.**".**

[Figure]

Figure R1. The intercomparison of PM$_{2.5}$ observations from U.S Department of State and from Beijing Municipal Environmental Protection Bureau (MEP)at sites close to each other (1.6 km) in 2014 – 2016.

*TP, RH, wspd and WD. PBLH was extracted from ECMWF interim re-analysis data. This is a very comprehensive suite of experimental data. All of these are obtained at different times and with different spatial scales and, for ground-based measurements, at different locations. Hence the data sets need to be colocated in both space and time before any analysis can be undertaken. Although locations indicated in section 2 are within 1 degree, an inhomogeneous megacity environment, both as regards surface characteristics and aerosol sources, may result in substantial spatial variation of the aerosol characteristics. These points need to be addressed in the MS.*

**In principle, we agree with the reviewer that aerosol properties could vary a lot with location, particularly in a megacity. This is one contributing factor to the AOD-PM$_{2.5}$ relationship as indicated in this study. In general, to do a statistical analysis between aerosol properties and meteorology, we need make the datasets collocated in both time and space. Unfortunately, it is challenging to exactly collocated them in most cases.**

**This study actually uses the TP RH, wspd and WD from the MEP site observation, which is about 1.6 km away from the aerosol observation site. Differences in these meteorology (particularly near surface) do exist and cause uncertainties to our study, but we believe this is the best we can do at this moment. For the PBL, it is extracted from ECMWF reanalysis data which has a horizontal resolution of around 12.5 km, and may be different from that at the aerosol observation site. However, considering that PBLH varies slowly with space and the observation of PBLH generally has an uncertainty larger than 100 m, we think the PBLH from ECMWF could be a good estimate to that at aerosol observation site. Moreover, an early study by Guo et al. (2016) compared the site**

observation of PBLH with ECMWF grid PBLH, which show good agreement. Of course, we agree with the reviewer that uncertainties could be introduced due to the mismatch of locations for these measurements. We have added partial description to indicate this in Pg8 L14-16: "We should admit that extra uncertainties could exist due to the distances between the MEP sit, U.S. Department of State site, and ECMWF grid, while they are close to each other.".

*Overall, the manuscript shows the influence of various factor on the PM$_{2.5}$/AOD relationship, but presentation lacks clarity and many things are left unexplained as detailed in my comments below. I miss clear conclusions (only a summary is provided at the end) and how the findings can be combined to provide a relationship between PM$_{2.5}$ and AOD, if that would be possible even for a single site.*

**We feel sorry that we have not delivered our conclusion clearly and have made careful revisions based on the reviewer's comments as listed below. We have also made a little more description about our conclusions, while the build-up of a specific relationship between PM$_{2.5}$ and AOD is still not aimed or reached in this study. The addition in the summary part lies in Pg 31 L17-21: "**With these findings, we need consider at least the impacts of PBLH, RH, Wind speed and wind direction, and use the AOD within PBL heights to build up better PM$_{2.5}$-AOD relationship. The impacts of these influential factors have been investigated while an optimal empirical PM$_{2.5}$-AOD relationship scheme has not been reached, which definitely need further study in future.**".**

*In general, the figures would be more interesting if they were easier to read, in some cases the text is too small. Furthermore, although a comparison is made between MODIS and AERONET data, in the main part of the MS only AERONET and CALIPSO data are sued. Hence I do not understand why MODIS is included.*

**These are good questions. The figures have been modified in order for readers to easily read and follow.**

**We included MODIS in our intercomparison study based on the consideration that significant uncertainties could exist in the satellite observations of AOD which can cause errors in the derived PM$_{2.5}$-AOD relationship: using AOD from different sources, such as ground-based and satellite based observations, or even different satellite observations, we could have different PM$_{2.5}$-AOD relationships due to their different AOD observations.**

*Detailed comments*

*p4, lines 2-3 specify modifying: scattering and absorption?*
**Corrected**

*5, 3 usually it's not the extinction coefficient that is provided but the AOD, i.e. the integrated extinction over the whole column*

In principle, many passive remote sensing instruments such as MODIS do provide AOD only rather than the extinction coefficient. However, the profile of mean extinction coefficient is being provided by some satellite products including CALIPSO level 2 aerosol profile products. This is why we provided the description here. We have slightly modified it in Pg 5 L1-4 as "Remote sensing observation generally provides the aerosol optical properties such as AOD and aerosol extinction coefficient, but not the aerosol mass or number concentration".

*5, 10 spelling van Donkelaar (also elsewhere in the MS)*
**Corrected**

*5, 11-13 these statements need some references*
**Three references have been added to support these statements, which are** (Paciorek et al., 2008; Li et al., 2017; Wang et al., 2017).

*6, 18 what do you mean with 'and so on'? Either specify or remove*
**It has been changed in Pg6 L17-18: "**The data used in this study are described as follows, including the data sources, their spatial and time resolutions, and the data period".

*8, 11 Was AERONET version 2 or the new version 3 used?*
**We chose to use AERONET version 2 product rather than the new version 3, because AERONET version 3 inversion algorithm was in development and the data were not released before our study.**

**Actually, AERONET version 3 has not been widely used until now. It can only provide us with the level 1.5 AOD on the website (https://aeronet.gsfc.nasa.gov/new_web/units.html), which are automatically cloud cleared but may not have final calibration applied. These data are not quality assured. Also, some other parameters including single scattering albedo (SSA), absorption and extinction aerosol optical depth cannot be acquired now. AERONET version 2 has been used in many studies and the differences in retrieved quantities between versions 2 and 3 are expected to be smaller than between versions 1 and 2 (Sayer A M et al. 2014). This is another reason that we used AERONET version 2.**

Reference: Sayer A M, Hsu N C, Eck T F, et al. AERONET-based models of smoke-dominated aerosol near source regions and transported over oceans, and implications for satellite retrievals of aerosol optical depth[J]. Atmospheric Chemistry and Physics, 2014, 14(20): 11493-11523.

*9, 8 No, AOD is not extinction multiplied by layer depth, but the integral of the extinction over the whole layer. This is different when extinction is not constant with height, as is usually the case.*
**We agree with the reviewer that the AOD is the integral of the extinction over the**

whole layer instead of a simple multiplication. CALIPSO level 2 aerosol profile products supply the extinction coefficient at different heights with a resolution of 60 or 120m in the vertical resolution. AOD at each layer is approximatively derived as the multiplication of the extinction coefficient and layer depth. Even though, we have modified the descriptions in Pg 10 L1-2 as "AOD at each layer is derived as the integration of the extinction coefficient within that layer".

*9, 20 why was MODIS C5.1 used and not the newer C6?*
**We tried our best to use MODIS C6 instead of C5.1 at our initial study. However, it is very challenging for the new C6 MODIS data to get downloaded. After trying many times, we gave up. Instead, we kept using MODIS C5.1 by only selecting those data with "very good" quality.**

*10, 2 why was Deep Blue selected and not dark target? Or the merged DB/DT AOD product from C6?*
**Deep Blue instead of dark target was selected for AOD product because the AOD accuracy is highly dependent on the surface reflectance for dark target product. The merged DB/DT AOD product was not used simply because we didn't obtain the C6 product.**

*10, 14 is it the optical thickness per unit mass concentration, or inversely the mass concentration per unit AOD, as eq 1 says?*
**We thank the reviewer for helping pointing this mistake out. We have corrected it on Pg 11 L9-10**: "where $\eta$ ($\mu g/m^3$) indicates the near surface aerosol $PM_{2.5}$ mass concentration per unit aerosol optical thickness".

*10, 16-19 it is not clear what extinction capability means: extinction per unit mass, or mass extinction coefficient? Or extinction efficiency? Or ....? Is it really true that for the same $PM_{2.5}$, the extinction is weaker for the same AOD? Or is it more complicated and does size distribution and RH have an influence?*
**We are sorry for the confusion. The extinction capability in this study actually means the mass extinction coefficient, we have explained this in Pg 11 L14-15**: "Note that the extinction capability here denotes the aerosol mass extinction coefficient".

**No, we do not mean that for the same $PM_{2.5}$, the extinction is weak for the same AOD. Instead, we believe that for the same $PM_{2.5}$, the larger the AOD, the larger the aerosol mass extinction coefficient which is defined as extinction capability here. Of course, if we go further into details, the aerosol extinction capability is definitely complicated which is dependent on multiple factors including size distribution and RH. These are what we have described in Pg 11 L10-15**: "Its value depends on the aerosol type, aerosol size, RH, PBLH, and the vertical structure of aerosol distribution. At the same $PM_{2.5}$ mass concentration, the smaller the AOD, the weaker the extinction capability; and the larger the AOD, the stronger the

extinction capability. Note that the extinction capability here denotes the aerosol mass extinction coefficient".

*11, 3 discrepancies or differences?*
**It should be 'differences" and we have corrected it.**

*11, 5 FMF is the fraction of the AOD due to fine particles (smaller than 1 micrometer); AE is exponent for the power law describing the wavelength dependence of the AOD*
**Many thanks to the reviewer for helping provide more accurate description. They have been modified as suggested.**

*11, 9 SSA is the ratio of the scattering coefficient to the extinction coefficient (=scattering + absorption)*
**It has been modified.**

*11, 12 replace According to with Following Lee et al. (2010)*
**Corrected**

*12, 2 what do you mean with "since the high percentage of"?*
  **This is a good question. According to the classification method proposed by Lee et al. (2010), aerosols can be classified into six types based on fine mode fraction (FMF) and single scattering albedo (SSA). Fine absorbing aerosols (SSA<=0.95, FMF>0.6 and AE>1.2) constitute the largest proportion among all the types in Beijing, which are 36.5%, 42.6%, 51.1% and 60.3% in four seasons respectively.**
  **To further study the absorptive properties of aerosols, fine absorbing aerosols are divided into heavily (SSA<=0.85), moderately (0.85<SSA<=0.9) and slightly (0.9<SSA<=0.95) absorbing aerosols in this study.**
  **Even with these considerations, using "since the high percentage of …" might be not suitable. We have removed this sentence and changed our classification categories to eight types instead of six types as shown in Pg 12 L10-21.**

*12, 6 replace "the method" with "the classification method"*
**Corrected**

*13, 6 what does it mean when MODIS correlated best, considering the bias, i.e. considering that MODIS AOD is too high?*
*Tables 1 and 2: what parameter is listed in the first 2 columns? How is bias calculated?*
**This is a good question. It simply means that the correlations between MODIS and AERONET are better in spring and summer than in other seasons, without**

**considering any bias. To be more accurate, we have modified the description in Pg 14 L20- Pg15 L7**: "The squared correlation ($R^2$) between MODIS and AERONET in Beijing are 0.81, 0.87, 0.69 and 0.34 in four seasons, of which the corresponding RMSE are 0.23, 0.29, 0.15 and 0.08. Low correlation in winter may be caused by the shortage of data samples compared to other seasons. When AOD becomes small, the relative errors in AOD from both MODIS and AERONET become large, which may cause the correlation of AOD between MODIS and AERONET also decrease as demonstrated in Table 1.".

**Table 1 and 2 compared paired AERONET and MODIS AOD, AERONET and CALIPSO in four seasons. Based on the satellite overpass time, the corresponding AERONET AOD averaged in time bins of within 30-min are compared to MODIS AOD and CALIPSO AOD respectively. The parameter in the first column is the averaged AERONET AOD, and the parameter in the second column is the averaged MODIS AOD (CALIPSO AOD) in Table 1 (Table 2). We have added the information into the Tables and described in Pg 14 L9-12**: "Table 1 further shows the inter-comparison results of AOD between AERONET and MODIS in spring (MAM), summer (JJA), fall (SON) and winter (DJF), which include their seasonal averaged AOD, squared correlation, absolute bias, relative bias and sample number"

*Bias* **in Table 1 is calculated as:**

$$Bias = MODIS \ AOD_{averaged} - AERONET \ AOD_{averaged}$$

*Bias* **in Table 2 is calculated as:**

$$Bias = CALIPSO \ AOD_{averaged} - AERONET \ AOD_{averaged}$$

**These calculation information have been added into Pg 14 L12-14**: "The absolute bias is calculated as the difference of seasonally averaged AOD from AERONET and MODIS at the same time; and the relative bias is calculated as the ratio of the absolute bias to the seasonally averaged AERONET AOD." **and Pg 15 L9-10**: "The bias shown in Table 2 is calculated in the same way as that in Table 1.".

*14, 1 suggest to change to: the hygroscopic growth factor f/(RH) is defined as the ratio ... ;*
**Corrected**

*Is it really the hygroscopic growth factor? This factor would relate to particle size, which indeed is one of the underlying factors, in addition to refractive index, influencing scattering; however, this article is about AOD, which is scattering and thus the scattering enhancement factor should be considered, see for instance Zieger et al., (2015) (Low hygroscopic scattering enhancement of boreal aerosol and the implications for a columnar optical closure study, Atmos. Chem. Phys., 15, 7247-7267, doi:10.5194/acp-15-7247-2015, 2015) and references therein for a*

*discussion on the subject and the f(RH) for different aerosol types in Europe.*

The reviewer proposed good question. The hygroscopic growth of aerosols with increasing relative humidity (RH) affects aerosol direct radiative effects by changing aerosol optical properties due to increasing water uptake of hydrophilic compositions such as sulfate, nitrate, and some organic matters. The aerosol hygroscopicity varies by space and time because of different aerosol sources, types, and chemical components (Li J et al, 2014).

Therefore, the hygroscopic growth parameterization of different aerosol types are also different as the reviewer commented. For example, the hygroscopic growth factor of sulfate optical properties is expressed as a function of relative humidity (RH) in an exponential way:

$$f(RH) = \exp[c_1 + c_2 / (RH + c_3) + c_4 / (RH + c_5)]$$

Where *f(RH)* denotes the hygroscopic growth factor of aerosol scattering coefficient, $c_1$ to $c_5$ are five wavelength dependent fitting coefficients taken directly from the CCM3 radiation package. The values of these coefficients are 11.24, 0.304, 1.088, 177.6 and 15.37, respectively, at a wavelength of 550 nm.

However, the scheme of Im et al. (2001) can be applied to other aerosol types (such as carbonaceous aerosols, etc.), which is in a form of power law:

$$f(RH) = (1 - RH)^{-g}$$

Where *g* is an empirical fitting value, setting to 0.38.

Zieger et al., (2015) tried to study the the magnitude of the scattering enhancement factor *f(RH)* in the boreal forest region of northern Europe. Also, they compared AOD achieved from ground-based in situ and remote sensing aerosol measurements

In fact, the hygroscopic growth correction is unable to be perfectly achieved using only one parameterization formula. Nevertheless, due to the lack of the observations of chemical composition in Beijing, in our study, *f*(RH) is expressed in a simple function as Eq 2, which is also used in Li C et al, (2005). We have also changed our description from "… is defined as …" to "can be defined as …".

*are retrieval errors. Since in this MS CALIPSO AOD is the main source for analysis, should a similar discussion be made for CALIPSO?*

**We are sorry that we did not deliver the information clearly and caused confusion. The description here is a general information that the hygroscopic growth process can cause a larger value of AOD compared to dry conditions, not for the comparison between MODIS and AERONET, or between CALIPSO and AERONET, since all of them are affected by this process. We have modified the description to make our point more clear in Pg 16 L6-12: "**The hygroscopic growth process has a significant contribution to AOD. Since $PM_{2.5}$ is often measured at a dry condition (< 40% in relative humidity), we often need consider the impacts of relative humidity to AOD in order to get a more reliable $PM_{2.5}$-AOD relationship. A dehydration adjusment can be applied to get the dry condition AOD, which is …**"**

*14, 13 what is the consequence of this assumption? Uniform mixing would imply that dry aerosol particles, water vapour and potential temperature are well-mixed, but RH would in that case increase with height and thus all aerosol parameters that vary with RH. In addition, often a scale height is used to account for an aerosol gradient.*

**The reviewer proposed a very good question. It is true that the RH would have large variation with height – would increase with height. This could definitely affect the dehydration adjustment of AOD. Currently, we only use the surface RH to do the adjustment which could cause the dry condition AOD is actually somehow overestimated compared its true value. We have removed the assumption and added this discussion into our manuscript in Pg 17 L4-7: "**In the atmosphere, the RH often increases with height within PBLH. This could definitely affect the dehydration adjustment of AOD in Eq. (3). Currently, we only use the surface RH to do the adjustment which could cause the dry condition AOD is actually somehow overestimated compared to its true value. **"**

*14, 19 this assumption implies that there are no disconnected aerosol layers and thus may introduce errors in experimental conditions where these may occur, as revealed by lidar.*

**This is a good point. The assumption made here does imply that there are no disconnected aerosol layers and could introduce errors in experimental conditions. We have carried out 4 different field observations in east China region and found for most time, the $PM_{2.5}$ mass concentration varies little with height within PBLH. Liu et al. (2009) also have the similar findings based on their 17 in-situ aircraft measurements in spring of 2005 and 2006 over Beijing. Therefore, our assumption could be valid for most cases. Even though, we agree with the reviewer and have added a discussion in Pg 17 L10-13: "**The calculation of column $PM_{2.5}$ mass concentration in Eq. (4) has implied that there are no disconnected aerosol layers and could introduce errors in experimental conditions, which was not considered in this study.**"**

**We agree with the reviewer and have made changes to our descriptions in Pg 17 L13-16:** "Eqs. (3) and (4) imply that for given PM$_{2.5}$, the increase of RH can result in the increase of AOD and the decrease of $\eta$, and that for given AOD, the increase of PBLH can cause the decrease of near-surface PM$_{2.5}$ concentrations and the decrease of $\eta$."

*15, 7 opposite trend, suggest to replace with anti-correlated;*
**Changed.**

*What are the colour bands in Fig 4a?*
**The colour bands in Fig. 4a are simply used to help readers see the anti-correlated temporal trends between PBLH and RH. The blue bands are for high PBLH and low RH, and the purple bands are for low PBLH and high PBLH, both of which indicate anti-correlated trends between PBLH and RH. Differently, the green (yellow) bands are for low (high) PBLH and low (high) RH, which indicates correlated trends of PBLH and RH. Clearly, there is generally an anti-correlated temporal trend between PBLH and RH. We have added the description about the colour bands into Pg17 L19-Pg18 L3:** "In Fig. 4a, the blue bands are for high PBLH and low RH, and the purple bands are for low PBLH and high PBLH, both of which indicate anti-correlated trends between PBLH and RH. Differently, the green (yellow) bands are for low (high) PBLH and low (high) RH, which indicates correlated trends of PBLH and RH. Clearly, there is generally an anti-correlated temporal trend between PBLH and RH." **The colour bands have also been explained in the caption of Figure 4.**

*15, 11 I assume that this is a typo and fig 4a is meant, but still don't understand the sentence*
**Yes, this is a typo and it is actually Fig. 4a. We are sorry that this sentence is kind of confusing. The main information we would like to deliver is that the effects of PBLH and RH should be considered when we study the PM₂.₅-AOD relationship. We deleted this sentence. Instead, we added the following sentence in Pg 18, L10-14:** "Without considering the variations of sources and sinks, PBLH is negatively correlated with PM$_{2.5}$, and RH is positively correlated with AOD. The anti-correlated trend between PBLH and RH shown in Fig. 4a imply that the effects of PBLH and RH on the PM$_{2.5}$-AOD relationship could be partially canceled out. However, it is still necessary to consider the effects of PBLH and RH for the study of PM$_{2.5}$-AOD relationship."

*16, top para: here AOD$_{dry}$ is plotted where AOD$_{dry}$ was obtained using the correction factor given by eq 2, which has to be explained. However, as mentioned above, RH is not constant with height, and hence f(RH) also varies with height. How was this accounted for? It seems that the good correlation between AOD$_{dry}$ and PM$_{2.5}$ is somewhat fortuitous. Does this good agreement lead to the conclusion that PM$_{2.5}$ was measured dry (at low RH)? See also the discussions on vertical variation of aerosol profiles later in this MS. Figs 5 and 6 both show the diurnal variation of PBLH and RH, for different seasons averaged over many years, for the whole day in Fig. 5, and for 5-20 in Figure 6 (a-d). I would expect that the curves for each season are exactly the same for the overlapping time periods. Could the authors explain the differences?*

**The reviewers proposed good questions and here are our explanations.**

(1) **Yes, we have added the explanation about the AOD$_{dry}$ and discussed the potential issues in our adjustment due to potential errors in RH, which are in Pg 19 L12-14**: "Note that the AOD$_{dry}$ is adjusted based on surface RH using Eqs. (2) and (3) and the vertical variation of RH has not been considered. As indicated earlier, the AOD$_{dry}$ obtained here could be somehow overestimated compared to its true value.".
**The impacts that we do not consider the vertical variation of RH has also been replied earlier and indicated in Pg 17 L4-7**: "In the atmosphere, the RH often increases with height within PBLH. This could definitely affect the dehydration adjustment of AOD in Eq. (3). Currently, we only use the surface RH to do the adjustment which could cause the dry condition AOD is actually somehow overestimated compared to its true value."

(2) **"PM$_{2.5}$ was measured at low RH" is not a conclusion, but a fact. The air samples before enter by the PM$_{2.5}$ instruments undergo a dry process (RH<35%), which ensure that all PM$_{2.5}$ observations are obtained at dry condition. We have added this information into the data part in Pg 7 L9-11: "**The air parcels before enter by the PM2.5 instruments undergo a dry process (RH<35%), which ensure that all PM2.5 observations are obtained at dry condition.**"**

**(3) While both Figs. 5 and 6(a) show the diurnal variation of PBLH and RH, they are for different data samples. What Figure 6 shows in the first row is the diurnal variation of multi-year averaged RH and PBLH when all measurements of RH, PBLH, AOD, PM$_{2.5}$ are available. By contrast, Figure 5 shows the variation once RH and PBLH are available.**

*17, 4-8 why is a linear relationship expected between AOD and PM$_{2.5}$, while the authors showed already in the above that there is not a good correlation, as also shown in Figs 6a2-d2? Why are these plots shown, why not only the corrected AOD$_{dry}$, since the necessity of that correction was discussed above? Furthermore, Figs 6a3-d3 do not show the R$^2$, they show time series which show that part of the time AOD$_{dry}$ and PM$_{2.5}$ trace each other, but at other times not; furthermore, these plots show the offset, i.e. AOD$_{dry}$ much larger than PM$_{2.5}$, which is in contrast to what was shown in Figure 4c, where the time series are much closer. Could the authors explain the differences? The PM$_{2.5}$ scales in Figs 4c and Fig 6 a3-d3 are much different but plotting on the same scales would make the discrepancy even larger.*

**These are very good comments and we highly appreciate them.**

(1)

[Figure]

**The Extinction Law (differential form)**

$$k_m(v) = -\frac{dI_v}{I_v \rho ds} = -\frac{dI_v}{I_v dM}$$

**Where $k_m(v)$ is mass extinction coefficient [$m^2 \times kg^{-1}$ ],**
**If $k_m$ keeps constant along the light path s, we have**

$$\tau_s(v) = k_m(v) \int_0^s \rho ds' = k_m(v)M$$

**Based on above equation, we could expect a linear relationship between AOD and aerosol mass concentration.**

**(2) The purpose that we show these figures is to see if the correction of PBLH and RH can improve the PM$_{2.5}$-AOD relationship, or a further examination of the effect of PBLH and RH, by comparing the results shown in Fig. 6a2-d2 and in Fig. 6a3-d3. AOD$_{dry}$ and PM$_{2.5\_column}$ have been shown in Fig. 6a3-d3, then we can examine their differences from those shown in Fig. 6a2-d2.**

**(3) Figs. 6a3-d3 do show the diurnal variation of the two examined variables, instead of scattering plots. The purpose is to see if they follow similar diurnal variation since both PBLH and RH have clear diurnal variation and the purpose of these figures is to examine the effect of PBLH and RH as indicated above. Even so, the squared correlation $R^2$ has been calculated and provided in the figure titles, which are 0.93, 0.84, 0.91 and 0.93 in four seasons respectively.**

**(4) There are offsets for the plots shown in Fig. 6. However, the two variables (AOD$_{dry}$ and PM$_{2.5}$) are using different y-coordinates, left and right respectively. Moreover, they have different yrange values. Furthermore, the data (so aerosol types and meteorology conditions) used in Figure 4c and the sub panels in Figure**

**6 are different (month in a particular year vs seasonal average). All of these make it challenging to compare the results in Figure 4c and Figure 6 directly. Our purpose is also not for this kind of intercomparison. The much closer time series shown in Figure 4c is simply due to the yrange selections of AOD and PM2.5. We could change the PM2.5 scales in Figure 6 to make AOD and PM2.5 lines closer. However, as said, the different data considered in Figure 4c and sub panels of Figure 6 make them hard to compare with each other.**

*17, 10-12 why only this narrow time frame (11-17 LT)? why does this limit the diurnal and seasonal variation on PM2.5/AOD?*

**The narrow time frame adopted in this study is not used to limit the seasonal variation, but to limit the diurnal variation on PM2.5/AOD. The reason is as follows. Atmospheric boundary layer generally increases from the morning, reaching the maximum at ~14:00 LT and decreases hereafter. Figure 5 in the manuscript shows that the PBLH is high around noon time and have weak changes. Also, RH has a low but weakly variable values also around noon time. So, we choose the time period between 11:00 and 17:00 local time to reduce the variability of PBLH and RH. By doing this way, we can guarantee a certain amount of data samples, and limit the influence of diurnal variation of PBLH and RH to PM2.5/AOD. We have modified/corrected our description to make it more accurate in Pg 21 L4-8:** "For this time period, the PBLH (RH) has high (low) values with weak variation, which make the impacts of PBLH and RH vary weakly with selected sample time in a season. By doing this, we try to keep a certain amount of data samples and limit the influence of diurnal variation of RH and PBLH on $\eta$."

*17, 13 Is the classification based on the scheme described on p. 11? Please refer to that scheme, and if not provide a reference. In Figure 7 are 8 aerosol types are shown whereas on p. 11 there are only 6, so I suspect that the some other classification than that on p. 11 is used*

**Yes, the aerosol classification in section 3.2 is bsed on the scheme described on pg 11. The scheme actually classifies the aerosol into 8 instead of 6 types in this study. We have corrected our description in Pg 11 to show the classification methods in Pg 12 L10-21**: "In this study, hourly averaged level 2 inversion products from AERONET at sites in Beijing are used, including FMF, AE and SSA data. Following Lee et al. (2010), aerosol is classified into eight types as follows: 1) Coarse non-absorbing (SSA>0.95, FMF<=0.4 and AE<=0.6), 2) Coarse absorbing (SSA<=0.95, FMF<=0.4 and AE<=0.6), 3) Mixed non-absorbing (SSA>0.95, 0.4<=FMF<0.6 and 0.6<=AE<1.2), 4) Mixed absorbing (SSA<=0.95, 0.4<=FMF<0.6 and 0.6<=AE<1.2), 5) Fine non-absorbing (SSA>0.95, FMF>0.6 and AE>1.2), 6) Fine highly-absorbing (SSA<=0.85, FMF>0.6 and AE>1.2), 7) Fine moderately-absorbing (0.85<=SSA<0.9, FMF>0.6 and AE>1.2), Fine slightly-absorbing (0.9<=SSA<0.95, FMF>0.6 and AE>1.2)."

*18, 5 what is 'heavily non-absorbing aerosols'? This type neither occurs in Fig 7 nor*

*on p. 11.*

*Figures 7 and 8 present the same aerosol types, but with a different colour scheme. It would be much easier to read when the colour schemes are the same. Please change.*

**The "heavily non-absorbing aerosols' is a typo, we have corrected it as 'fine non-absorbing aerosol' in Pg 22 L1. Following the great suggestion, we have also modified Figure 8 to make its color scheme as the same as that used in Figure 7.**

*18, 14 extinction capacity? On p.10 extinction capability was used. Is that different? I don't understand either of these terms, see my comment on p. 10.*

**Yes, this is the same as that shown in Pg 10, which is the mass extinction coefficient. We have indicated this in the manuscript in Pg 11 L14-15: "Note that the extinction capability here denotes the aerosol mass extinction coefficient."**

*18, Fig 8 and Table 3 discussion: rather than only providing correlation coefficients, the authors should discuss the reasons for the observed differences. There are two fundamental reasons for the variation of the $PM_{2.5}$/AOD ratio: $PM_{2.5}$ does not include coarse particles while they do contribute to the extinction at wavelengths in the visible, and thus to AOD. In addition, the extinction efficiency peaks for a particle size which depends on the ratio between particle size and the wavelength.*

**Great comments and we highly appreciate the reviewer's help. Following this suggestion, we have modified our descriptions in Pg 22 L11-Pg23 L1**: "Theoretically, aerosol extinction capacity increases with particle size parameter $(x=2\pi r/\lambda)$ and reaches a maximum value when size parameter is around 6. Therefore, for solar visible radiation (such as $\lambda$=500 nm), the extinction capacity for aerosol particles generally increases with size for particles with radius less than 0.5 μm, and then decreases when radius larger than 0.5 μm. Actually, for the wavelength of 550 nm, the extinction efficiency of fine-mode particles (peak radius ranging from ~0.11 to ~0.33 μm) is stronger than coarse-mode aerosols. Moreover, coarse particles, which may be not included in $PM_{2.5}$, can contribute a lot to the extinction at wavelengths in the visible, and thus to AOD. This is especially true for dust days dominated by coarse-mode aerosols, of which high AOD is more likely to be due to $PM_{10}$ rather than $PM_{2.5}$. These make the lower $\eta$ for coarse-mode than fine mode aerosol."

*19, 1-5 Figure 9 is too small to read the numbers, but it is clear that there are large differences in the slopes from one aerosol type to another, as also stated on line 9. Hence I don't understand where the numbers given on line 3 come from.*

**We have modified Figure 9 so that the numbers in the figure can be easier to read. The numbers given on line 3 comes from our analysis which is not shown in the manuscript. As shown in Figure R2, we carried out the linear regression analysis for all types of aerosols. The slopes of the linear regression functions (PM₂.₅=a×AOD+b) are 90.16, 56.9, 117.97 and 138.42 in four seasons respectively. We have modified our description in Pg23 L9-12: "We have also done the linear regression analysis for all types of aerosols which is not shown here, and found that the slopes of the linear regression functions (PM₂.₅=a×AOD+b) are 90.16, 56.9,**

117.97 and 138.42 in four seasons respectively."

[Figure]

**Figure R2.** Scatterplots of AERONET AOD vs. PM$_{2.5}$ concentrations of different aerosol types in four seasons for the period of 2011 to 2015 in Beijing.

*19, 6 why does high RH increase the extinction efficiency? Since the refractive index of water is lower than that of most other materials, would the condensation of water vapour lower the extinction efficiency?*

**The reviewer proposed a good question. The mass extinction coefficient of particles is dependent on the refractive index of the mater and its size parameter. While the refractive index of water is lower than that of most other materials, the hygroscopic growth of aerosol particles under high RH condition makes more particles with size parameters more close to 5 or 6, in other words, makes more particles with strong extinction coefficient. Thus, the high RH can increase the extinction capacity of aerosol.**

*19, 10 Similar as on line 3, I do not understand where these numbers come from. I cannot find them in Fig 9*

**These numbers are in the linear fitting regression equations which are shown in Figure 9 for different types of aerosols in four seasons. We have added them into Figure 9 and made modification to the figure and descriptions in Pg 23 L20 – Pg24 L13:** "However, there are large differences in the slope of regression functions among different aerosol types. For absorbing aerosols, the slope roughly decreases with increasing particle size from coarse, mixed to fine particles, with values of about 89, 111, 104 μg/m³ in spring, 85, 122, 74 μg/m³ in summer, 71, 163, 131 μg/m³ in fall, and 44, 143, 158 μg/m³ in winter. The slope is also generally larger for absorbing than non-absorbing aerosol. The slopes for mixed absorbing and non-absorbing aerosol are

111 and 65 μg/m$^3$ in spring, 122 and 40 μg/m$^3$ in summer, 163 and 109 μg/m$^3$ in fall, and 143 and 89 μg/m$^3$ in winter. And the slopes for fine absorbing and non-absorbing aerosol are 105 and 76 μg/m3 in spring, 74 and 65 μg/m$^3$ in summer, 131 and 96 μg/m$^3$ in fall, and 158 and 122 μg/m$^3$ in winter. Thus, same as shown in Fig. 8, the slope roughly decreases with particle size, with small values for coarse-mode aerosols and large values for fine-mode aerosols in four seasons, and the slope of non-absorbing aerosols is generally smaller than absorbing aerosols. "

*19, 12 Figure 4 does not show that the slope decreases with particle size.*
**Many thanks to the reviewer. This is a typo, it is actually Figure 8 instead of Figure 4. We have corrected it.**

*19, 15 this conclusion is clear, if only looking at Figure 9. However, I do not understand how the discussion in this section supports this conclusion.*
**We agree with the reviewer and made changes to our discussion to support this conclusion in Pg 23 L20 to Pg 24 L13:** "However, there are large differences in the slope of regression functions among different aerosol types. For absorbing aerosols, the slope roughly decreases with increasing particle size from coarse, mixed to fine particles, with values of about 89, 111, 104 μg/m$^3$ in spring, 85, 122, 74 μg/m$^3$ in summer, 71, 163, 131 μg/m$^3$ in fall, and 44, 143, 158 μg/m$^3$ in winter. The slope is also generally larger for absorbing than non-absorbing aerosol. The slopes for mixed absorbing and non-absorbing aerosol are 111 and 65 μg/m$^3$ in spring, 122 and 40 μg/m$^3$ in summer, 163 and 109 μg/m$^3$ in fall, and 143 and 89 μg/m$^3$ in winter. And the slopes for fine absorbing and non-absorbing aerosol are 105 and 76 μg/m3 in spring, 74 and 65 μg/m$^3$ in summer, 131 and 96 μg/m$^3$ in fall, and 158 and 122 μg/m$^3$ in winter. Thus, same as shown in Fig. 8, the slope roughly decreases with particle size, with small values for coarse-mode aerosols and large values for fine-mode aerosols in four seasons, and the slope of non-absorbing aerosols is generally smaller than absorbing aerosols."

*20, 13-15 this is a very surprising conclusion, regarding the location of Beijing with near-by mountains blocking the circulation, pollution advected form the south and clean air from the north, as well as the transport of desert dust aerosol in the spring. What leads to this conclusion? I don't see it in Figure 10. And also the first 5 lines on top of p. 2 contradict this conclusion.*
**We highly agree with the reviewer. We made a more careful examination and found that conclusion is erroneous. We corrected them with more analysis in Pg 25 L11- Pg26 L2**: "Different from the wind speed which will be analyzed in Figs. 11 and 12, the influence of wind direction to the AOD-PM$_{2.5}$ relationship is often combined with the effect of wind speed. Beijing is surrounded by Hebei province and mountains in the northern areas. When the winds come from south, Beijing is in the downstream location to the pollution source from Hebei and the pollutants could be further accumulated in Beijing due to the mountain blocking effect. By contrast, when the winds come from north, Beijing is in the upstream region relative to the pollution

source in Hebei, and the cold air from north can disperse the air pollutants. As shown in Figure 11, with similar wind speed, the occurrence rate of heavy air pollution is much higher for cases with winds from the south than from the north. Moreover, the aerosol pollution events also decrease with increasing wind speed for cases with winds both from the north and the south."

*20, Fig 11 does not show haze occurrence but AOD and PM$_{2.5}$.*
**We agree with the reviewer that Figure 11 actually shows AOD and PM$_{2.5}$ instead of haze occurrence. We have modified our description in Pg 26 L3 – L13:** "Figure 12 illustrates the relationship between the severity extent of aerosol amount denoted by AOD and PM$_{2.5}$ and surface wind speed. For good air quality with PM$_{2.5}$<50 μg/m$^3$, the occurrence rate increases with increasing wind speed, ranging from 39.3% (v<=1 m/s) to 92.9% (v>7 m/s). Differently, the occurrence of poor air quality with PM$_{2.5}$>150 μg/m$^3$ ranges from 20.92% (v<=1 m/s) to 0 (v>7 m/s). The weakening of surface wind speed reduces the transport of near-surface aerosol to the outside regions, leading to the build-up and continuance of heavy aerosol pollution condition in Beijing. On the contrary, the increase of surface wind speed, which may be due to the development of weather system like monsoon in Beijing, causes the disperse of aerosols, and then reduction of the heavy aerosol pollution occurrence rate."

*22, 2 and fig 13: what is actually plotted: the axis annotation reads AOD, but AOD thus far was the column integrated extinction, i.e. independent of height. May be extinction is plotted? Should these profiles reflect that the PBLH varies with season, and thus also the aerosol vertical distribution vary with season? The strong decrease with height contradicts the assumption of a uniform mixing (p. 14).*
**The reviewer proposed very good questions here. Actually, what Figure 13 shows is similar to the extinction, instead of the columnar AOD. More correctly, the AOD in figure 13 is the integrated extinction at each atmospheric layer. In CALIPSO level 2 product, AOD_layer and AODcolumn could be calculated as**

$$\begin{cases} AOD_h = k_h \times \Delta h \\ AOD_{column} = \sum_{h=0}^{H} AOD_h \end{cases}$$

**Where $AOD_h$ represents AOD at the altitude of $h$, $k_h$ is the extinction coefficient at the altitude of $h$, $\Delta h$ is layer depth and $AOD_{column}$ is the AOD of the whole column.**

**We agree with the reviewer that the profile in Figure 13 actually partially reflect that the that PBLH varies with season. Moreover, the layer integrated values also depend on the aerosol type and size distribution in addition to PM$_{2.5}$, so it could vary with height within PBLH. We have removed Figure 13, 14 and corresponding descriptions in the revision version. By the way, the assumption of a uniform mixing in pg14 has been removed in this revision.**

*22, 9 as for Fig 13, I don't understand what is plotted here*
**Figure 14 has also been removed.**

*22, 18-21: same comments as for the top of the page. I don't understand what is shown here, and hence it is hard to comment. I also don't understand why in summer, with peak AOD at a height of 516 m, is concluded that most aerosol is below that height. Integration of the profiles below and above 500 m would give a number supporting this conclusion.*
**This figure and its corresponding descriptions have been removed. For your information, this figure is actually the probability density distribution function of layer integrated AOD with altitude in four seasons.**

*23, eq 5 This eq needs some more explanation. Apparently the authors assume here that CALIPSO AOD underestimates the true value (fig 3)? And therefore they scale to AERONET AOD?*
     **What the reviewer understands is right. The quality-assured level 2.0 AERONET AOD data with a stated uncertainty of 0.015 were used as the "ground truth" (Holben et al., 1998). Compared with AERONET AOD, there exist errors in CALIPSO AOD measurements, which might be caused by cloud and near-surface noise. Thus, we use the CALIPSO vertical profile product to obtain the proportion of AOD at each layer to AOD of the whole column in the vertical direction. Combined with Aeronet AOD, AOD of different altitude can be acquired as accurately as possible. We have add he explanation in Pg 29 L5- 8: "As shown in Figure 3, the CALIPSO seems underestimate AOD compared to AEORNET. We here treat the AERONET AOD as more reliable or "ground truth" data, and use the CALIPSO vertical profile to scale the AERONET AOD for its vertical distribution.".**

*23, 15 how is top of atmosphere defined? Figure 15 would be easier to understand when the AOD was plotted on the same scale. Note that, PM$_{2.5}$ is plotted vs AOD, not AOD vs PM$_{2.5}$ as the caption says.*
     **Here, the top of atmosphere is actually for the columnar atmosphere that the satellite can observe, which is about 705 km. We have modified it as "For heights of 500 m, 1000 m, PBLH and the whole atmospheric column, …" in Pg 29 L11-13.**
     **Figure 15 has been modified by following the reviewer's suggestion. The title has been corrected as PM$_{2.5}$ vs AOD.**

*24, 1-3 this conclusion is clear and follows form Figure 13, and hence Fig 15 is obsolete, including the explanation on p.23; it also shows that the uniform-mixing assumption on p 14. is not valid for aerosol.*
     **We agree with the reviewer. Currently, we have removed Figures 13 and 14 along with their descriptions. Instead, we kept using Figure 15 and its description.**

**The results shown in Figure 15 do indicate the varying extinction coefficient with heights within PBLH. This could be related with the changes of RH and particle size distribution with heights. We have removed the assumption used on Pg 14 by citing several aircraft observations which found that PM2.5 varies little with height within PBLH in Pg 16 L14-17: "**
[revised manuscript text omitted]

---

## Referee Report (RR1)

The authors present an interesting and comprehensive analysis of correlation between MODIS AOD and ground-based PM2.5 at Beijing. More importantly, influential factors affecting the PM-AOD relationship have been analyzed from the perspective of diurnal variation, which has the potential to shed light on the possibility to PM remote sensing from space. Overall, the paper is well written and I only have the following comments for the authors to address before its acceptance for publication in ACP.

**Major points to be considered:**

1. Section 2.1: The authors only depicted the use of 1:30 LT MODIS AOD data. They should better clarify the time of day for other data used here, including PM2.5, and related meteorological variables. In addition, the spatial and temporal averaging scheme they took should be clarified as well, given their large diurnal variability and spatial variation.

2. Figure 14: More discussion is necessary for the statement "the slopes of linear regression lines vary a lot for heights 500 m, 1000 m and PBLH, but much smaller for $H$ above PBLH". For instance, PBLH exhibits large diurnal variation, which is quite different from 500m, 1000m for different seasons. This will inevitably affect the correlation between AOD and PM2.5 and its slope.

**Minor points to be considered:**

1. Abstract: "atmospheric boundary layer height (PBLH)" -> " planetary boundary layer height (PBLH) "

2. Page 2 line 10: it is better to clarify the four seasons in "with aerosol type in four seasons respectively "

**3. Introduction:** With regard to the investigation of the relation between MODIS AOD and ground-based PM concentrations, several important references have been missing. For example, Guo et al. (2009) for the first time reported the correlation between MODIS AOD and ground-based PM1/PM2.5/PM10 across eastern China

based on long-term collocated MODIS AOD and hourly PM measurements from China Atmosphere Watch Network (CAWNET) of Chinese Meteorological Administration. They also discussed the potential influences of PBLH and RH on the correlation between PM and AOD. This CAN be added before " Xin et al. (2015) investigated the relationships .... " IN PAGE 5.

Reference:

Guo J.P., Zhang X.Y., Che H.Z., Gong S.L., An X.Q., Cao C.X., Guang J., Zhang H., Wang Y.Q., Zhang X.C., Zhao P., and Li X.W., 2009. Correlation between PM Concentrations and Aerosol Optical Depth in Eastern China, Atmospheric Environment, 43(37): 5876-5886.

4. Page 7, line   6-7: " U.S Department of State"-> "U.S. Embassy Beijing "

5. Page 7: line 14: the latitude and longitude for the MEP site should be given.

6. Page 8 line 1: " MEP.... away from the U.S. Department of State site are provided by CMA, ":   I am curious how the meteorological data observed at the site of MEP can be obtained from CMA?   It will be better if the authors can plot a map showing the locations of US Embassy Beijing, MEP site, and CMA site.

7. Page 9 line 6: Beijing (39ºN, 116ºE) - > " Beijing (39ºN, 116ºE) "

8. Page line 16: " sit" is a typo.

9. Page 10 line 4: " the AOD profiles " should be revised to "the aerosol extinction coefficient profile" or sth else because AOD is a notion of integration and can not be defined as a profile.

10. Page 19 lines 8-9: " is contributed " -> "could be attributed to"

11. Page 19 line 21: " increase and decrease " -> "increasing and decreasing"

12. Page 20 line 3: " After PBLH and RH corrections " -> "After being corrected for

PBLH and RH"

13. Page 20 line 10: " weak variation, which make " - > " weak temporal variation, which makes"?

14. Page 26 line 9: " in 532 nm band " - " at 532 nm band "

15. Page 27, line 13: " but varies " -> "and varies"

16. Figure 2 cannot be seen clearly. Cautions should be taken when insert a figure. What do the red dash lines in Figure 14 mean? It will be better to give some necessary description in the caption of Figure 14.

---

## Author Response (AR2)

Response to Reviewer #4

We thank the reviewer for the detailed and valuable comments and suggestions that have helped us improve the paper. Our detailed responses (**Bold**) to the reviewers' questions and comments (*Italic*) are listed below.

*Review of Zheng et al., "Analysis of influential factors for the relationship between PM2.5 and AOD in Beijing" submitted for publication in ACP.*

*General comments*
*The authors investigate the relationship between AOD and PM2.5 using 5 years of data in Beijing, figuring out the impacts of different influential factors to this relationship including PBLH, RH, wind speed and direction and aerosol type. This is a complex study, which found valuable and interesting quantitative results regarding the impacts of most influential factors to AOD-PM2.5 relationship.*

*Relationship between PM2.5 and aerosol optical depth (AOD) is often investigated in order to obtain surface PM2.5 from satellite observation of AOD with a broad area coverage. Various factors could affect the AOD-PM2.5 regressions, but they have not yet been systematically evaluated. This study evaluates the influence of most various factors to the AOD-PM2.5 relationship quantitatively, and thus could have a broad influence. This will help future derivation of accurate PM2.5 from satellite AOD observations. The authors have done a very good reply to the previous reviewers' comments. The paper is also written well, but some language corrections/improvements are needed as detailed below. I would recommend its publication after a minor revision.*
**We highly appreciate the reviewer's evaluation about the value of our study and the detailed comments which help us improve the paper quality.**

*Detailed Comments*
*1. Many "aerosols" in the manuscript could be changed as "aerosol".*
**We agree with the reviewer and have made thorough changes accordingly.**
*2. Page 4, line 8, "are associated with" should be "is associated with"*
**Corrected.**
*3. Page 5, line 5, 'observations' -> 'observation'*
**Corrected.**
*4. Page 5, line 11, 'has not' should be changed as "has no" or "has not a"*
**It has been corrected as "has no".**
*5. Page 6, lines 9 and 18, "includes" should be changed to "include"*
**Corrected.**
*6. Page 7, line 11, 'ensure' should be 'ensures'*
**Corrected.**
*7. Page 8, "RH" and "PBLH" have been denoted earlier.*
**Only abbreviation words are provided here now.**
*8. Page 8, please make sure whether you are using "PBLH" or "BLH" for planetary*

*boundary layer height and keep them consistent.*

**Many thanks for this comment. We have modified them all as "PBLH" for consistency.**

*9. Page 9, line 5, change 'are' to 'is'*

**Corrected.**

*10. Page 10, line 2, change 'is' to 'are'*

**Corrected.**

*11. Page 10, line 15, change 'is' to 'are'*

**Corrected.**

*12. Page 11, line 4, delete 'has'*

**Done.**

*13. Page 11, line 6, 'PM2.5' to 'PM$_{2.5}$'*

**Corrected.**

*14. Page 13, line 4, 'including those from both MODIS and CALIPSO'*

**Corrected.**

*15. Page 13, line 6, 'AODs at time within 30-min frame'*

**Corrected. Also corrected a few other similar issues.**

*16. Page 14, line 2, 'AOD' -> 'AODs'*

**Corrected.**

*17. Page 14, line 6, 'substantial' -> 'large'*

**Changed.**

*18. Page 16, line 1, 'observational studies'*

**Changed.**

*19. Page 16, line 11, please use 'PBLH' for consistency.*

**Changed.**

*20. Page 17, line 6-9, you could modify the sentence as "Clearly, there is generally an anti-correlated temporal trend between PBLH and RH. The averaged PBLHs for 2011 to 2015 are 2.56 km, 1.97 km, 1.55 km and 1.32 km with corresponding averaged RHs of 27.58%, 48.73%, 42.78% and 33.05% in MAM, JJA, SON and DJF, respectively"*

**Modified (Pg 18 L17-Pg 19 L1).**

*21. Page 17, line 12, 'imply' -> 'implies'*

**Corrected.**

*22. Page 17, line 19, 'trend' -> 'trends'*

**Corrected.**

*23. Page 18, line 18, delete 'promising'*

**Deleted.**

*24. Page 19, lines 6-10, the descriptions could be further improved.*

**We have modified this sentence in Pg 20 L20-Pg21 L2 as: "**In terms of diurnal variation, it shows that from 8:00 to 14:00 LT, the solar radiation that surface receives increases, making PBLH rise and RH decrease gradually. It also shows that PBLH at 14:00 LT is the highest and RH at 14:00 LT is the lowest within the whole day.**".**

*25. Page 19, line 21, 'R$^2$ values"*
**Changed.**
*26. Page 21, line 8, change 'decrease' to 'decreases'*
**Corrected.**
*27. Page 22, line 9, using 'and thus not shown here".*
**Corrected.**
*28. Page 23, line 11, 'implies'->'imply'*
**Corrected.**
*29. Page 24, line 8, please delete "Different from the wind speed which will be analyzed in Figs. 12 and 13"*
**Deleted.**
*30. Page 26, line 16, 'distribution also'*
**Corrected.**
*31. Page 26, line 17, 'ADO integrated from surface to'*
**Corrected.**
*32. Page 28, line 3, 'AOD' changes to 'AODs'*
**Corrected.**
*33. Page 28, lines 5 and 7, 'a RMS' -> 'an RMS'*
**Corrected.**
*34. Page 28, line 12, I would suggest "With the correction of RH and PBLH to AOD, R2 of monthly averaged PM2.5 and AOD increases from 0.63 to 0.76 at 14:00 LT, ..."*
**Modified.**
*35. Page 29, lines 2-4, "The occurrence rate of good air quality (PM2.5<50 µg/m3) increases ..."*
**Modified.**
*36. The figures are in good quality.*
**Thanks!**

Response to Reviewer #5

We thank the reviewer for the detailed and valuable comments and suggestions that have helped us improve the paper. Our detailed responses (**Bold**) to the reviewers' questions and comments (*Italic*) are listed below.

*The authors present an interesting and comprehensive analysis of correlation between MODIS AOD and ground-based PM2.5 at Beijing. More importantly, influential factors affecting the PM-AOD relationship have been analyzed from the perspective of diurnal variation, which has the potential to shed light on the possibility to PM remote sensing from space. Overall, the paper is well written and I only have the following comments for the authors to address before its acceptance for publication in ACP.*

**We highly appreciate the reviewer's evaluation about the value of our study and the detailed comments which help us improve the paper quality.**

*Major points to be considered:*

*1. Section 2.1: The authors only depicted the use of 1:30 LT MODIS AOD data. They should better clarify the time of day for other data used here, including $PM_{2.5}$, and related meteorological variables. In addition, the spatial and temporal averaging scheme they took should be clarified as well, given their large diurnal variability and spatial variation.*

**The reviewer proposed good comments. The time information regarding other data have been added in the data part, which are either hourly or 3-hourly datasets.**

**The spatial and temporal averaging scheme has been clarified now in Pg 7 L8-17:** "For comparisons of surface PM2.5 and satellite AOD, the hourly surface PM2.5 mass concentrations around the satellite overpass time and the instant satellite AOD or aerosol profiles at the grid (5 km resolution for CALIPSO and 10 km resolution for MODIS) closest to the surface site, have been used. For the influential analysis to the surface $PM_{2.5}$ and AERONET AOD relationship, hourly averaged data of PM2.5, AOD, and meteorological variables at CMA site (e.g., PBLH, RH and winds) have been adopted. For time without PBLH observations from CMA radiosonde profiles, the 3-hourly PBLH from ERA reanalysis has been interpolated at the grid close to CMA site."

*2. Figure 14: More discussion is necessary for the statement "the slopes of linear regression lines vary a lot for heights 500 m, 1000 m and PBLH, but much smaller for H above PBLH". For instance, PBLH exhibits large diurnal variation, which is quite different from 500m, 1000m for different seasons. This will inevitably affect the correlation between AOD and $PM_{2.5}$ and its slope.*

**We agree with the reviewer that PBLH generally has large diurnal variation and considerable seasonal variation. We here do not intend to use 500 m or 1000 m as the PBLH, but would like to indicate that the variability of AOD-$PM_{2.5}$ relationship to the heights used in the AOD integration. Following this suggestion, we added the information as discussion in Pg 29 L14-17: "PBLH generally has**

large diurnal variation and considerable seasonal variation, which is quite different from 500 m and 1000 m for different seasons. This will inevitably affect the correlation between AOD and PM2.5 and its slope.**"**

*Minor points to be considered:*

*1. Abstract: "atmospheric boundary layer height (PBLH)" -> "planetary boundary layer height (PBLH)"*

**Corrected.**

*2, Page 2 line 10: it is better to clarify the four seasons in "with aerosol type in four seasons respectively"*

**Corrected. We have modified it to** "It shows that $\eta$ varies from 54.32 to 183.14, 87.32 to 104.79, 95.13 to 163.52 and 1.23 to 235.08 μg/m$^3$ with aerosol type in spring, summer, fall and winter respectively." **in Pg 2 L10.**

*3. Introduction: With regard to the investigation of the relation between MODIS AOD and ground-based PM concentrations, several important references have been missing. For example, Guo et al. (2009) for the first time reported the correlation between MODIS AOD and ground-based PM$_1$/PM$_{2.5}$/PM$_{10}$ across eastern China based on long-term collocated MODIS AOD and hourly PM measurements from China Atmosphere Watch Network (CAWNET) of Chinese Meteorological Administration. They also discussed the potential influences of PBLH and RH on the correlation between PM and AOD. This CAN be added before "Xin et al. (2015) investigated the relationships ...." IN PAGE 5.*

**We appreciate the reviewer's comments and have modified the figure to make it more clear in the new manuscript. Meanwhile, we have added necessary description as the reviewer suggested in the caption of Figure 14**: "
[revised manuscript text omitted]

---

## Author Response (AR3)

Many thanks to the editor for these helpful technical corrections. Following the comments (*Italic*), we have made corresponding corrections (**Bold**) as listed below.

*The manuscript has been much improved with the revision. I appreciate the efforts the authors put into revising their manuscript. I still have several technical corrections before publishing. Please make the corrections:*

*1. Pg2, L19, change "to" to "for".*
**Changed.**

*2. Pg4, L10. Since you are talking about aerosol acting as ice nuclei, please add some references on this aspect such as:*
*Hoose, C. and Möhler, O.: Heterogeneous ice nucleation on atmospheric aerosols: a review of results from laboratory experiments, Atmos. Chem. Phys., 12, 9817–9854, doi:10.5194/acp-12-9817-2012, 2012.*

*Liu, X., Shi, X., Zhang, K., Jensen, E. J., Gettelman, A., Barahona, D., Nenes, A., and Lawson, P.: Sensitivity studies of dust ice nuclei effect on cirrus clouds with the Community Atmosphere Model CAM5, Atmos. Chem. Phys., 12, 12061–12079, doi:10.5194/acp-12-12061-2012, 2012.*
**We agree with the editor and have added more references as suggested.**

*3. Pg5, L15. Change "building" to "development".*
**Changed.**

*4. Pg6, L16. The definitions of acronyms "PBLH" and "RH" should be done earlier since they are used on Page 5.*
**Yes, they have been defined in page 5 now. Thanks.**

*5. Pg7, L9. change "instant" to "instantaneous".*
**Changed.**

*6. Pg8, L6. change "enter by" to "entering".*
**Changed.**

*7. Pg9, L10. change "MEP" to "CMA".*
**Corrected.**

*8. Pg12, L17. remove "the" before "AOD".*
**Removed.**

*9. Pg17, L15 and L16. "Within PBLH" should be "within PBL". You need to spell out PBL.*
**We agree with the reviewer. The PBLH in Lines 6,7,8,10,15 and 16 in Pg 17 and**

**L20 in Pg 28 are all corrected.**

*10. Pg20, L7. change "contributed" to "attributed".*
**Corrected.**

*11. Pg24, L3 and L4. "For absorbing aerosols, the slope roughly decreases with increasing particle size from coarse, mixed to fine particles..." however, is the particle size decreased instead of increased from coarse, mixed to fine particles? the slope does not decrease ( 89, 111, 104). This sentence needs to be rewritten for correction.*
**Many thanks for helping figure this error here. We have changed the sentence as "For absorbing aerosols, the slope roughly increases with decreasing particle size from coarse to mixed particles, with values of about 89, 111 µg/m3 in spring, 85, 122 µg/m3 in summer, 71, 163 µg/m3 in fall, and 44, 143 µg/m3 in winter, respectively."**